# Deep mutational scanning quantifies DNA binding and predicts clinical outcomes of PAX6 variants

Alexander F McDonnell [ID], Marcin Plech [ID], Benjamin J Livesey, Lukas Gerasimavicius [ID], Liusaidh J Owen, Hildegard Nikki Hall, David R FitzPatrick [ID], Joseph A Marsh [ID] & Grzegorz Kudla [ID] ✉

## Abstract

**Nonsense and missense mutations in the transcription factor PAX6 cause a wide range of eye development defects, including aniridia, microphthalmia and coloboma. To understand how changes of PAX6:DNA binding cause these phenotypes, we combined saturation mutagenesis of the paired domain of PAX6 with a yeast one-hybrid (Y1H) assay in which expression of a PAX6-GAL4 fusion gene drives antibiotic resistance. We quantified binding of more than 2700 single amino-acid variants to two DNA sequence elements. Mutations in DNA-facing residues of the N-terminal subdomain and linker region were most detrimental, as were mutations to prolines and to negatively charged residues. Many variants caused sequence-specific molecular gain-of-function effects, including variants in position 71 that increased binding to the LE9 enhancer but decreased binding to a SELEX-derived binding site. In the absence of antibiotic selection, variants that retained DNA binding slowed yeast growth, likely because such variants perturbed the yeast transcriptome. Benchmarking against known patient variants and applying ACMG/AMP guidelines to variant classification, we obtained supporting-to-moderate evidence that 977 variants are likely pathogenic and 1306 are likely benign. Our analysis shows that most pathogenic mutations in the paired domain of PAX6 can be explained simply by the effects of these mutations on PAX6:DNA association, and establishes Y1H as a generalisable assay for the interpretation of variant effects in transcription factors.**

**Keywords** Deep Mutational Scanning; Transcription Factor; Eye Development
**Subject Category** Chromatin, Transcription & Genomics

See also: AF Rubin

## Introduction

Transcription factors (TFs) are one of the largest human gene families with ~1600 genes (Lambert et al, 2018), and are key for development, signal transduction and cellular housekeeping (Karin

and Smeal, 1992; Brivanlou and Darnell, 2002; Zabidi et al, 2015). They act by selectively binding DNA elements and facilitating or inhibiting transcription machinery recruitment onto the promoter regions of target genes, modulating their expression (Spitz and Furlong, 2012). Many TFs sit atop gene regulatory networks and function as master regulators of development by coordinating cellular processes that include stem cell renewal, lineage specification, and patterning (Lambert et al, 2018; Spitz and Furlong, 2012; Lee and Young, 2013; Thompson et al, 2021). Mutations that disrupt the ability of TFs to bind DNA are commonly associated with disease (Lee and Young, 2013; Chi and Epstein, 2002).

PAX6 is a highly conserved transcription factor essential for the correct development of the central nervous system, the pancreas, and the eye. It belongs to the Pax (paired box) family of transcription factors and consists of an N-terminal paired domain, a central homeodomain and a C-terminal proline/serine/threonine-rich domain (Cvekl and Callaerts, 2017; Dohrmann et al, 2000; Manuel et al, 2015; Ochi et al, 2022). In humans, PAX6 is first expressed in the surface and neural ectoderm through the activity of a tightly orchestrated array of enhancer elements, and it induces the expression of multiple downstream genes that collectively regulate the balance between neural progenitor cell self-renewal and differentiation (Terzic and Saraga-Babic, 1999; Sun et al, 2015; Sansom et al, 2009; Thakurela et al, 2016; Aota et al, 2003; Bhatia et al, 2013; Inoue et al, 2007; Hayashi et al, 1987; Narasimhan et al, 2015; Epstein et al, 1994; Xu et al, 1999; Coutinho et al, 2011). Studies using Hidden Markov models and chromatin immunoprecipitation (ChIP) experiments confirm that PAX6 binds to hundreds of genomic loci associated with embryonic development, patterning, and transcriptional regulation (Sun et al, 2015; Coutinho et al, 2011). The pivotal role of PAX6 in oculogenesis was famously highlighted in early experiments where its misexpression resulted in the formation of ectopic eye structures in both vertebrates and invertebrates (Chow et al, 1999; Halder et al, 1995).

Mutations in PAX6 cause a range of developmental disorders (reviewed here (Cunha et al, 2019)), with homozygous deletions being lethal in mice and humans, while heterozygous deletions, nonsense, and frameshift mutations typically cause aniridia in humans. By contrast, missense variants in the paired domain of PAX6 often produce other ocular pathologies, such as microphthalmia, anophthalmia, coloboma, and isolated foveal hypoplasia. Knowledge of the impact of PAX6 missense variants on binding to DNA is key for genetic diagnosis of eye development disorders

MRC Human Genetics Unit, Institute of Genetics and Cancer, University of Edinburgh, Edinburgh EH4 2XU, UK. ✉E-mail: gkudla@gmail.com

and can facilitate patient care (Kejun Tang et al, 1997; Williamson et al, 2019). However, exploring TF-DNA interactions has so far been limited by relatively low-throughput techniques that are resource-intensive and can only feasibly be performed on a handful of variants.

Deep Mutational Scanning (DMS) (Fowler et al, 2023; Hietpas et al, 2011; Wei and Li, 2023; Fowler and Fields, 2014; Tabet et al, 2022) is a high-throughput experimental strategy to map the effects of mutations in a gene on a selected cellular or molecular phenotype, such as cell growth rate, gene expression level, or protein interaction with a ligand. In DMS, a library of mutants is constructed and the phenotypes of all mutants are measured in parallel, typically in a pooled cell-based assay using a next-generation sequencing readout. DMS has been applied to classify disease mutations, predict the evolution of viruses, inform precision medicine approaches and study the mechanisms through which mutations influence function (Findlay et al, 2018; Starita et al, 2017; Starr et al, 2020). DMS has also been applied to human TFs (Giacomelli et al, 2018; Staller et al, 2018; Kitzman et al, 2015), including TP53. Most of these DMS assays used cell growth rate as phenotypic readout. However, only a minority of TFs have a measurable effect on growth in human cell models, making it difficult to generalise the TP53 experimental approach to other TFs.

We hypothesised that the functional effects of transcription factor mutations could be studied using an adaptation of a yeast one-hybrid (Y1H) assay. In Y1H, a TF binding site (the "bait") is cloned upstream of a reporter construct such as an antibiotic resistance gene that is expressed upon binding of a cognate TF (the "prey"), resulting in antibiotic resistance (Reece-Hoyes and Walhout, 2012). Y1H assays have typically been used to screen libraries of putative binding sites for their ability to interact with a TF of choice in TF-centred approaches (Ji et al, 2018), but they can also be used to screen libraries of TFs for the ability to bind a specific DNA element (gene-centred) (Reece-Hoyes et al, 2011). By applying the assay to a library of single amino acid variants of the PAX6 paired domain, we validate Y1H as a generally applicable modality for DMS assays, we identify positions in PAX6 with differential effects upon binding to different DNA sequence elements, and we build an accurate classifier of the clinical effects of mutations in PAX6.

## Results

### A high-throughput yeast one-hybrid assay to characterise *PAX6* variants

We designed an experimental strategy to systematically measure the effects of single amino acid mutations in the paired domain of human *PAX6* on binding to DNA (Fig. 1A). The key steps of our approach are: (1) the construction of a saturation mutagenesis library of PAX6; (2) expression of PAX6 variants in a yeast strain containing a PAX6-binding site upstream of an antibiotic resistance gene; (3) pooled competitive growth assays in the presence and absence of antibiotic; (4) high-throughput sequencing of samples isolated at multiple time points during competitive growth to associate functional scores with PAX6 variants.

We started by developing a Y1H assay in which expression of the PAX6 paired domain fused to the GAL4 activation domain

(henceforth collectively referred to as simply PAX6) conferred geneticin resistance to yeast (*S. cerevisiae*) cells. Geneticin resistance was mediated by binding of PAX6 to a DNA response element upstream of the Geneticin[R] gene integrated into the yeast genome. We tested 16 candidate PAX6 DNA response elements (Appendix Table S1, Appendix Fig. S1a), from which 2 that conferred specific antibiotic resistance upon PAX6 binding were chosen: LE9, a minimum PAX6-responsive element derived from the head surface ectoderm enhancer that autoregulates expression of *Pax6* in ectoderm derivatives in the eye (Aota et al, 2003), and BLX, a sequence derived from an unpublished SELEX screen for shared PAX6 paired domain and SOX2-binding sequences (Fig. 1B). The remaining 14 candidate DNA response elements were rejected because they did not offer antibiotic resistance (possibly due to insufficient copy numbers, low PAX6 affinity, or PAX6 binding requiring the presence of cofactors such as SOX2 (Inoue et al, 2007; Narasimhan et al, 2015; Kamachi et al, 2001)), or they resulted in antibiotic resistance in the absence of PAX6 (likely caused by interactions with endogenous yeast transcription factors). The CRY response element looked promising in the spot assay (Appendix Fig. S1a), but showed high antibiotic resistance in the absence of PAX6 in liquid media, and was excluded from analysis. Fusion of GAL4 activation domain at the C-terminus of the PAX6 paired domain conferred antibiotic resistance whereas an N-terminal fusion did not. We replicated these results in liquid culture assays, and confirmed that previously studied PAX6 disease variants that disrupt DNA binding did not generate antibiotic resistance (Williamson et al, 2019) (Appendix Fig. S1b, Dataset EV1).

Encouraged by these results, we used one-pot saturation mutagenesis (Wrenbeck et al, 2016) to generate a library of single amino acid variants of the PAX6 paired domain and its flanking sequence (amino acid positions 1–150; Fig. 1C). We then used the Gateway system to clone the variant library into a centromeric yeast expression plasmid library named pMP1_XL that included a strong synthetic promoter (Redden and Alper, 2015) and a collection of 30-nt barcodes upstream from the promoter. Phasing of PAX6 variants with their corresponding barcodes was done using PacBio sequencing of the plasmid library followed by analysis with alignparse (Crawford and Bloom, 2019). The final library contained 96.5% (2731/2831) of all possible single nonsynonymous (missense) variants, 99.3% (148/149) of all possible single stop codon (nonsense) variants, 227 synonymous variants, and 5266 unique variants with two or more mutations (Appendix Fig. S2). Missense and nonsense single variants were associated with a median of 19 barcodes per variant, and 8,655 barcodes were associated with wild-type PAX6.

### PAX6 variants cause sequence-specific perturbations in DNA-binding

To quantify DNA binding of PAX6 variants, we transformed the mutant library into LE9 and BLX yeast strains, treated cells with geneticin, collected samples at 12 h intervals, and quantified barcodes by Illumina sequencing (see Methods). Barcode counts were analysed with the Enrich2 pipeline (Rubin et al, 2017), yielding fitness scores for 95.8% (2856/2980) of variants in the LE9 assay and 95.2% (2838/2980) of variants in the BLX assay. Fitness scores were normalised so the average of wild-type variants was 0 in both assays. To quantify the reproducibility of these scores, we

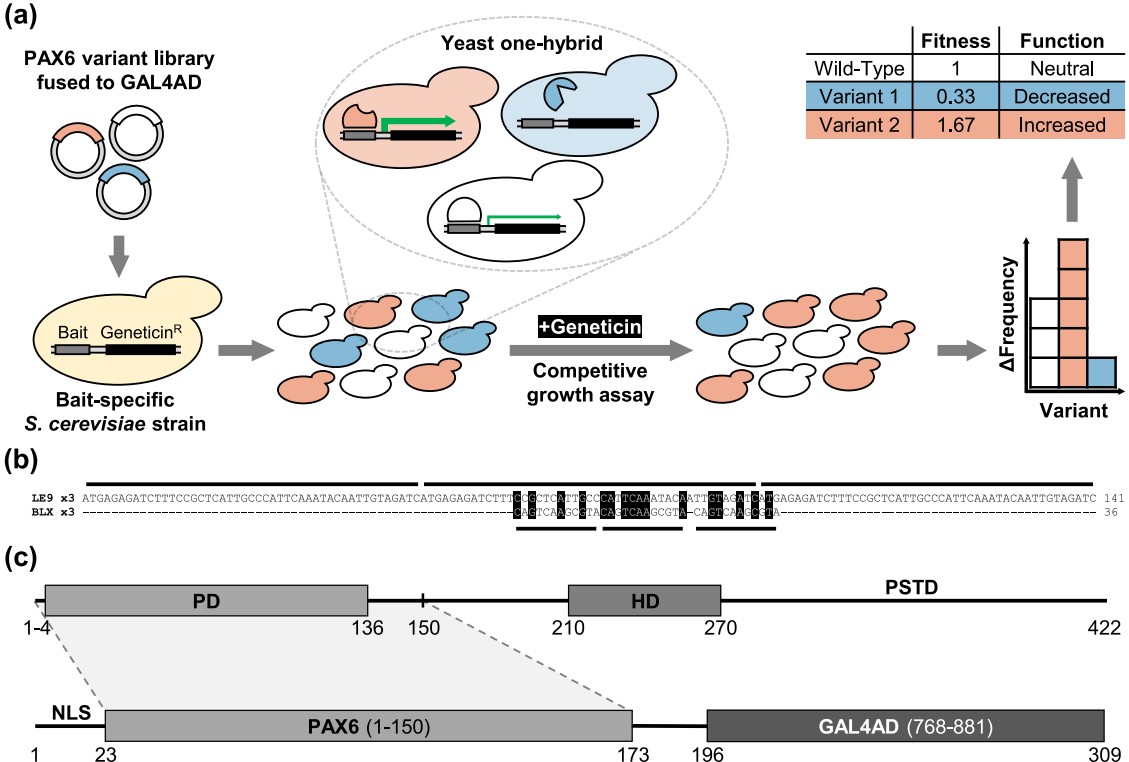

**Figure 1. Experimental design of the deep mutational scan of PAX6 paired domain.**

(A) A pool of barcoded variants of human PAX6 paired domain fused to the yeast GAL4 activation domain (GAL4AD) was transformed into yeast containing a specific PAX6 DNA response element (bait) upstream of an antibiotic resistance gene (Geneticin[R]). Transformants were grown together in a competitive growth assay in the presence of the antibiotic geneticin, such that PAX6 variants with disrupted DNA binding were depleted in the population and those with unperturbed DNA-binding were conversely enriched. Changes in variant frequency were measured by counting barcodes using next-generation sequencing, and fitness scores were subsequently calculated using Enrich2 (Rubin et al, 2017). (B) PAX6 bait sequences used in the DMS assays aligned using CLUSTALW. Each bait sequence consisted of three tandem repeats, indicated by horizontal broken lines. Sequence homology is highlighted in black. (C) (top) Linear structure of PAX6 depicting the paired domain (PD), homeodomain (HD) and proline-serine-threonine-rich transactivation domain (PSTD); (bottom) the PAX6-GAL4AD fusion protein constituting the "prey" component of the yeast one-hybrid assay. NLS, nuclear localisation sequence; PAX6 (1–150), PAX6 paired domain with short flanking sequence additions; GAL4AD; GAL4 activation domain.

analysed sets of barcodes associated with the same PAX6 variant. After separating all barcodes into two equal sets and calculating variant scores for each set, we obtained Pearson correlations of 0.77 for the LE9 assay and 0.63 for the BLX assay for all barcodes. As expected, reproducibility increased with the number of barcodes per variant and with read coverage per barcode (Appendix Fig. S3).

Variant fitness scores showed broadly similar patterns in the LE9 and BLX assays, with a Spearman correlation of 0.64 between assays (Figs. 2, EV1 and EV2). The BLX assay appeared more tolerant of PAX6 mutations (Fig. 2B,E), perhaps reflecting stronger binding of PAX6 to the SELEX-derived consensus binding site. Premature termination codons (PTC; nonsense) were detrimental throughout the gene in each assay. This was expected because a PTC in PAX6 paired domain results in a truncated protein that lacks GAL4 activation domain and should not induce antibiotic resistance. The mean scores of synonymous variants (LE9 mean = −0.09; BLX mean = −0.24) were slightly lower than wild-type PAX6, but this was most likely caused by a small number of outliers with a single barcode per variant, as the effect all but disappeared when variants with three or more barcodes were considered (Fig. 2D; Appendix Fig. S4). Most missense variants were

detrimental to function (Fig. 2B,F), with some positions appearing completely intolerant of amino acid substitutions (blue columns) while other positions tolerated substitutions well (white columns). Substitutions to proline and to negatively charged amino acids were most detrimental, consistent with their roles in disrupting secondary structure elements and electrostatic interactions with DNA, respectively. Mutational sensitivity was correlated with PAX6 structural elements (Fig. 2A) and with sequence conservation (Fig. 2C). The apparent periodicity in α-helices α1, α4 and α5 was reminiscent of α-helix periodicity of 3.6 residues per helical turn, suggesting that one face of the helix may be more sensitive to mutations than the other. Periodicity was also observed in the β1 and β2 beta sheets (Fig. 2B).

Some positions showed distinct patterns of mutational sensitivity: positions 50, 70, and 71 in the LE9 assay, and 50 and 54 in the BLX assay, were enriched for mutations that apparently increased DNA binding compared to wild-type PAX6 (Fig. 2B). Position 17 was unique as many mutations in this position reduced yeast fitness to a larger extent than PTCs or other presumed loss of function mutations; we hypothesise that mutations in position 17 generate PAX6 variants that are toxic to yeast through a molecular gain-of-

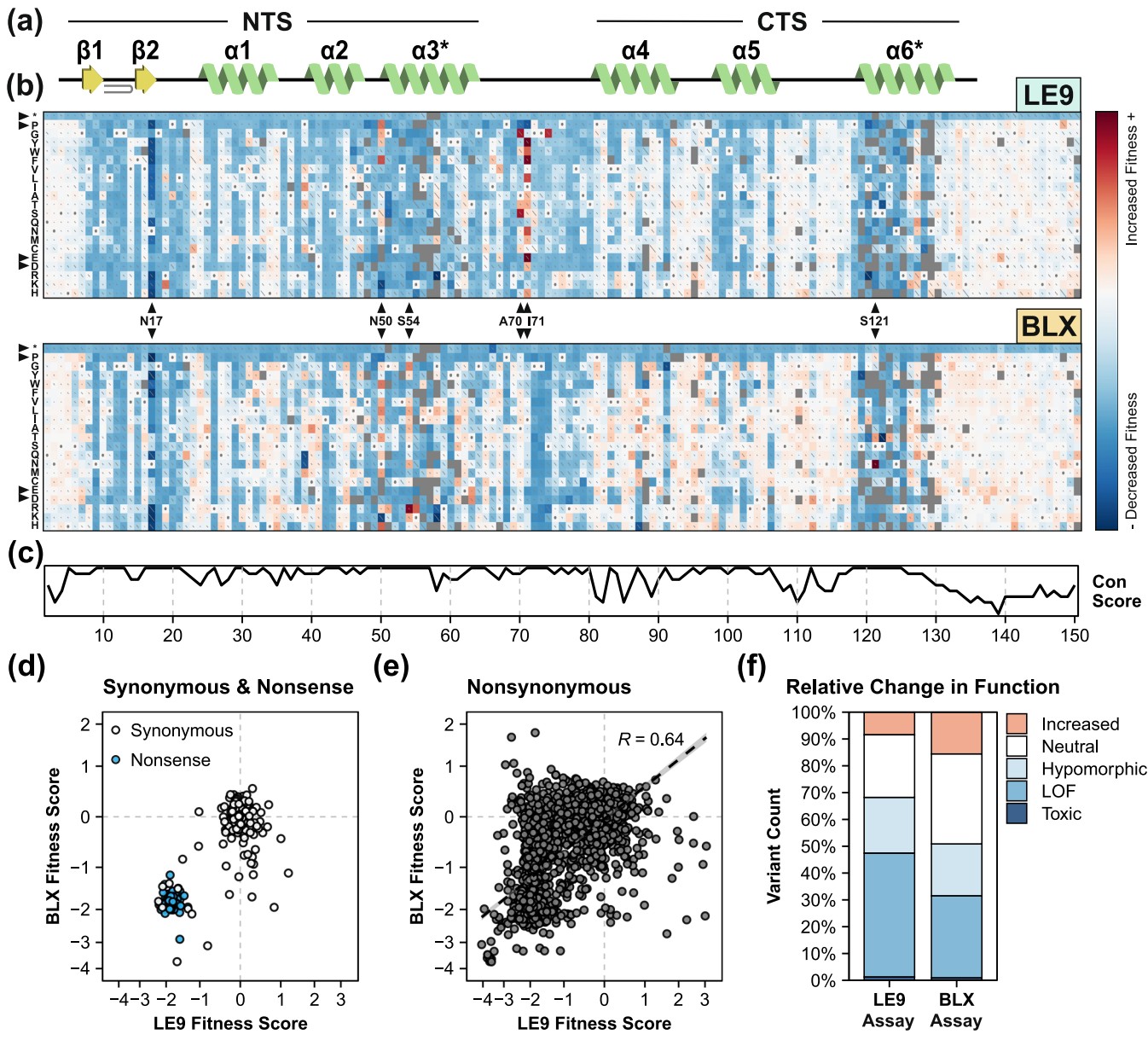

**Figure 2. Mutational Landscape of PAX6 paired domain.**

(A) Secondary structure of PAX6 paired domain depicting the N-terminal subdomain (NTS) and C-terminal subdomain (CTS) joined by a short flexible linker. α-helices and β-sheets are represented in green and yellow, respectively. A short β-turn is shown in the N-terminal portion of the NTS. Primary DNA-recognition helices in the NTS and CTS are denoted using asterisks (*). (B) Sequence-function maps generated following the deep mutational scan of PAX6 using LE9 (upper) and BLX (lower) as bait sequences in the presence of geneticin, with the position in PAX6 (x axis) and amino acid variant (y axis) shown. Each tile represents a PAX6 variant, with fill colour indicating an increased (positive, red) or decreased (negative, blue) fitness score relative to the wild-type (score of 0, white). Variants with missing data are shown as filled grey tiles. Grey dots within tiles depict the wild-type sequence at each position, and diagonal lines represent the standard error for the score, and are scaled such that a diagonal covering the entire square is the highest standard error on the plot. Highlighted are positions with highly distinct fitness scores across DMS assays (black triangles along the X axis) and substitutions to stop codons, prolines, and negatively charged amino acids (black triangles along the Y axis). (C) PAX6 evolutionary conservation score (Con score) generated using ConSurf (Ashkenazy et al, 2016). (D, E) Comparisons of fitness scores between the LE9 and BLX assays of synonymous (white) and nonsense (blue) variants (D), and all single nonsynonymous variants (E). R, Spearman correlation. Dashed lines represent the fitness score for wild-type PAX6. (F) Proportion of fitness scores in functional categories defined by comparison (FDR < 0.05) with wild-type and nonsense fitness score distributions (see Methods). Increased, increased function; Neutral, wild-type-like; Hypomorphic, intermediately perturbed function; LOF, complete loss of function; Toxic, toxic to yeast cells.

function mechanism. Certain variants appear to have opposed effects on fitness across the two assays—for example, N50R, G51R, S54R, S74G, S121N, and multiple substitutions at position I71 all increased the fitness score in one DMS assay (LE9 or BLX) while decreasing fitness in the other (Fig. 2B). Similarly, there were numerous variants such as C52G, A70S, I71L, G72S, and R128Q that showed either a significant change in fitness score or no effect on fitness, depending on the DNA response element. Analysis of

replicate barcodes in these variants suggested that these were genuine effects of PAX6 binding to different DNA response elements, rather than experimental noise (Appendix Fig. S5). All these variants are within 5 Å from the nearest non-hydrogen atom in DNA, consistent with a direct effect on PAX6:DNA interactions.

## Structural sensitivity of PAX6 to amino acid substitutions

Mutations in the interior of proteins or at protein:DNA interfaces are known to be more detrimental than mutations in surface residues (Livesey and Marsh, 2022a; Ferrer-Costa et al, 2002; King and Jukes, 1969). To verify that this principle applies to PAX6, we analysed the effects of PAX6 mutations in the context of the PAX6:DNA complex structure. We used the 6pax crystal structure from PDB to model the interactions of PAX6 paired domain with LE9 and BLX binding sites (Xu et al, 1999). As expected, we found greater effects of mutations in inward-facing residues of the subdomain protein cores and the protein:DNA interface (Fig. 3A,D,G). The proximity of each residue to DNA correlated with fitness score, with residues closest to DNA having the lowest median fitness scores (Fig. 3B,E).

Previous studies suggested that the N-terminal subdomain (NTS) had a larger impact on PAX6:DNA interactions, compared to the C-terminal subdomain (CTS) (Xu et al, 1999, 1995; Treisman et al, 1991). Our analysis confirms this, with NTS mutations being more consistently detrimental than CTS mutations or mutations outside the paired domain (Fig. 3C,F). Mutations in the linker sequence between the two subdomains were slightly less deleterious than NTS mutations in the LE9 assay but were more deleterious in the BLX assay. We speculate that the effects of mutations in the linker sequence depend on the relative arrangement of the binding sites of the NTS and CTS subdomains in the DNA structure. For example, a larger distance between these sites will favour linker mutations that promote a more extended conformation.

## PAX6 reduces yeast growth through promiscuous binding to the genome

While PAX6 allows yeast strains containing LE9 and BLX bait sequences to grow in geneticin-containing media, we noticed that expression of wild-type PAX6 caused a small but reproducible reduction of growth in the absence of antibiotic. To test whether this effect was specific to wild-type PAX6, we repeated the DMS assays with no antibiotic in the culture media (Figs. 4, EV3 and EV4). Intriguingly, we observed that wild-type PAX6 and variants that can bind DNA reduced growth in the absence of antibiotic, whereas loss-of-function variants showed relatively faster growth. In particular, amino-acid mutations in the PAX6:DNA binding interface prevented growth in media containing geneticin, but increased growth without geneticin (Fig. 4A–C) We observed negative correlations of variant effects when comparing DMS assays with and without geneticin (Fig. 4D,E. LE9 +geneticin vs -geneticin: R = −0.73; BLX +geneticin vs -geneticin: R = −0.54).

We considered several possible explanations for these observations: (1) expression of PAX6 may impose a metabolic load and divert resources from essential cellular pathways; (2) misfolding of mutant PAX6 may cause a cellular stress response; (3) gratuitous induction of the Geneticin[R] gene may impose a metabolic cost; (4)

non-specific binding of PAX6 to yeast genomic DNA might reduce growth by perturbing the yeast transcriptome. Scenarios (1) and (2) do not explain the results as they predict mutant PAX6 to have similar (1) or lower (2) growth rates than wild-type PAX6 in the absence of geneticin, whereas we observed an increased growth rate in PAX6 mutants. To distinguish between (3) and (4), we reasoned that cost related to Geneticin[R] gene expression should result in strong negative correlations within the same bait strain grown with and without antibiotic, whereas the cost related to non-specific genome binding of PAX6 should result in a strong positive correlation between LE9 and BLX strains grown without antibiotic. We found the strongest correlation (R = 0.95) between LE9 and BLX strains without antibiotic (Fig. 4F), suggesting that the identity of the bait sequence is not relevant in this condition. This analysis supports the idea that functional (DNA-interacting) variants of PAX6 reduce growth in the absence of geneticin through promiscuous binding to the yeast genome.

## DMS accurately classifies pathogenic variants and gives insight into disease phenotypes

Recent studies show the utility of deep mutational scanning in the annotation of variants of uncertain significance (VUS) (Findlay et al, 2018; Fayer et al, 2021). We therefore examined if PAX6 fitness scores could help explain variant pathogenicity. Putatively benign variants found in the gnomAD database had fitness scores similar to synonymous variants (Fig. 5A–D); this was more pronounced with than without geneticin (compare Fig. 5A–D). By contrast, pathogenic variants collected from the HGUeye, ClinVar and HGMD databases had a mean fitness score close to nonsense variants (Fig. 5A–D). While most variants followed these patterns, this was not absolute, with some pathogenic variants showing neutral or increased function relative to the wild-type, and some gnomAD variants demonstrating varying degrees of reduced function. Moreover, these effects were often DNA response element-specific, suggesting the observed effects on fitness were a result of PAX6:DNA binding, rather than stability-related disruption.

Deep mutational scanning has also been used to benchmark in-silico variant effect predictors (VEPs). We therefore compared the accuracy of classification of PAX6 disease mutations based on DMS and VEPs (Fig. 5E, Appendix Table S2). Raw PAX6 fitness scores classified benign and pathogenic variants better than most VEPs (LE9 ROC AUC: 0.885, BLX ROC AUC: 0.891). Previous studies indicated that the magnitude of change in function rather than the sign caused by amino acid substitution offers greater accuracy in classifying pathogenic variants (Livesey and Marsh, 2022a; Gerasimavicius et al, 2020). Accordingly, we calculated absolute values (ABS) for fitness scores and assessed the predictive performance (Fig. 5E). Interestingly, we saw an improvement in both DMS assays (LE9 ABS ROC AUC: 0.913, BLX ABS ROC AUC: 0.954), with BLX ABS fitness scores outperforming all VEPs. LE9 ABS fitness score outperformed all unsupervised VEPs, with only MetaRNN and MutPred surpassing it in ROC AUC (Li et al, 2022; Pejaver et al, 2020).

The observation that sequence-function maps with (Fig. 2B) and without (Fig. 4B) geneticin are nearly mirror images of each other prompted us to explore whether assays without geneticin accurately predict the pathogenicity of PAX6 mutations (Fig. 5E). Remarkably,

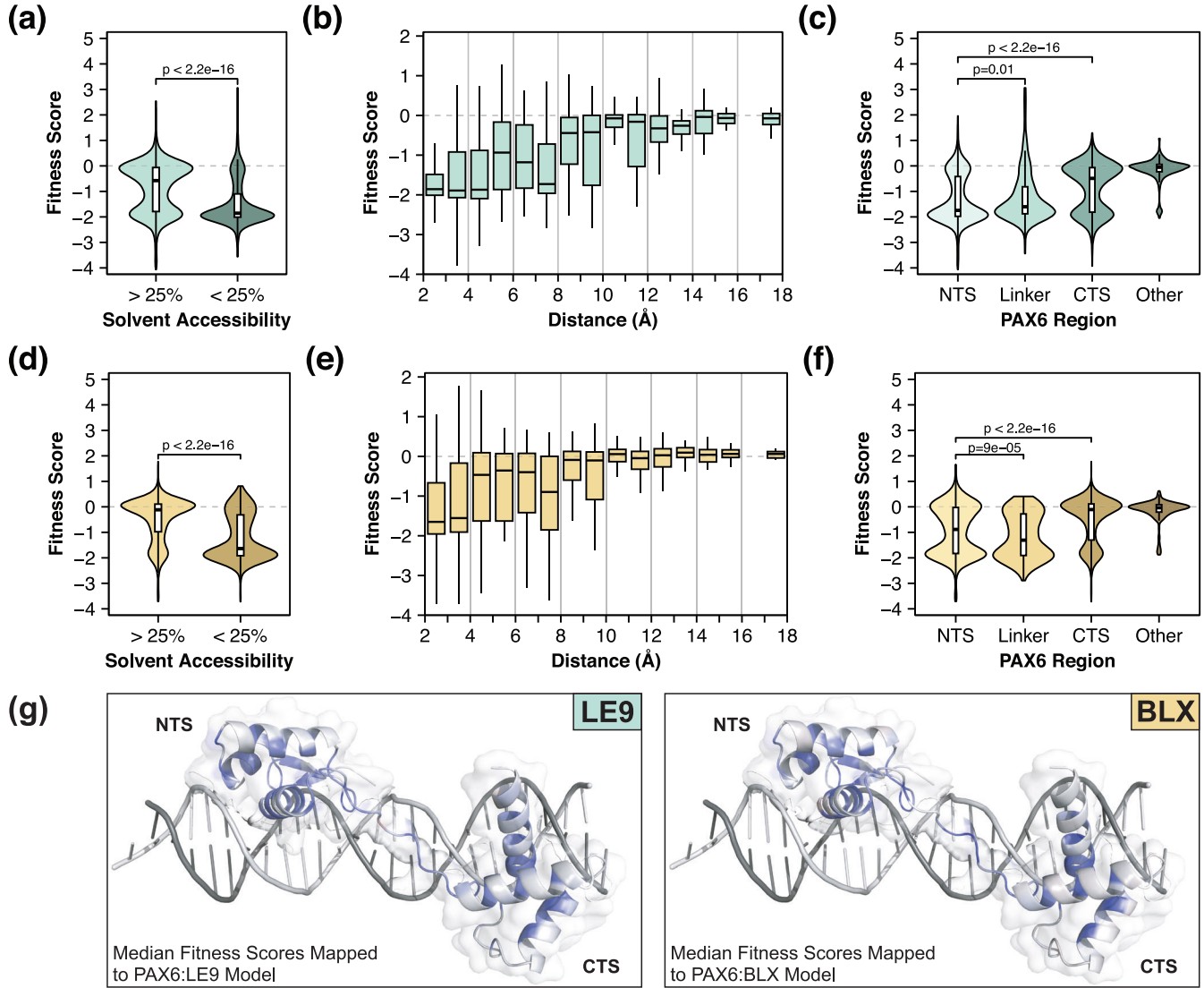

**Figure 3. Effects of mutations on protein structure.**

Distributions of variant fitness scores from the LE9 (A–C) and BLX (D–F) assays, grouped by solvent accessibility, distances from DNA, and region of PAX6. Numbers of variants are $n = 2856$ for LE9 and $n = 2838$ for BLX. Dashed lines represent the fitness score for wild-type PAX6. Solvent accessibility was defined using a threshold of 25% solvent accessible surface area. Distances between each amino acid residue and the nearest non-hydrogen atom in DNA were measured and discretised into 1 Å bins. PAX6 regions were defined as NTS (4–63), Linker (64–79), CTS (80–136) and 'other' (1–3 and 137–150) according to the 6pax crystal structure. In the box plots, boxes indicate the first and third quartiles, whiskers extend to the most extreme value within 1.5 times the inter-quartile range, and lines within boxes indicate medians. All p-values were calculated using the Wilcoxon test and adjusted using FDR. (G) PAX6:DNA structures were generated by modelling LE9 (left) or BLX (right) in place of the DNA sequence crystallised in the 6PAX crystal structure (Xu et al, 1999). Median fitness scores were calculated per residue and mapped onto each structure and coloured using a gradient where red and blue represent increased or decreased binding, respectively, and white as no change. NTS N-terminal subdomain, CTS C-terminal subdomain. Source data are available online for this figure.

the LE9 and BLX assays without geneticin offered greater predictive accuracy compared to their respective assays with geneticin (LE9 -geneticin ROC AUC: 0.892; BLX -geneticin ROC AUC: 0.898). Using absolute fitness scores improved accuracy further (LE9 ABS -geneticin ROC AUC: 0.922; BLX ABS -geneticin ROC AUC: 0.926), with both assays surpassing all other DMS assays and unsupervised VEPs in ROC AUC with the exception of BLX ABS with geneticin, and were only surpassed by the supervised VEPs MetaRNN and MutPred (Li et al, 2022; Pejaver et al, 2020).

We next used a proposed modification to American College of Medical Genetics (ACMG)/Association for Medical Pathology (AMP) guidelines for variant interpretation to quantify the evidence of pathogenicity/benignity generated in this study (Richards et al, 2015; Brnich et al, 2019). Our findings indicate that the BLX ABS assay provides moderate evidence of pathogenicity for 941 PAX6 variants, supporting evidence of pathogenicity for 36 variants, moderate evidence of benignity for 1184 variants and supporting evidence of benignity for 122 variants. The

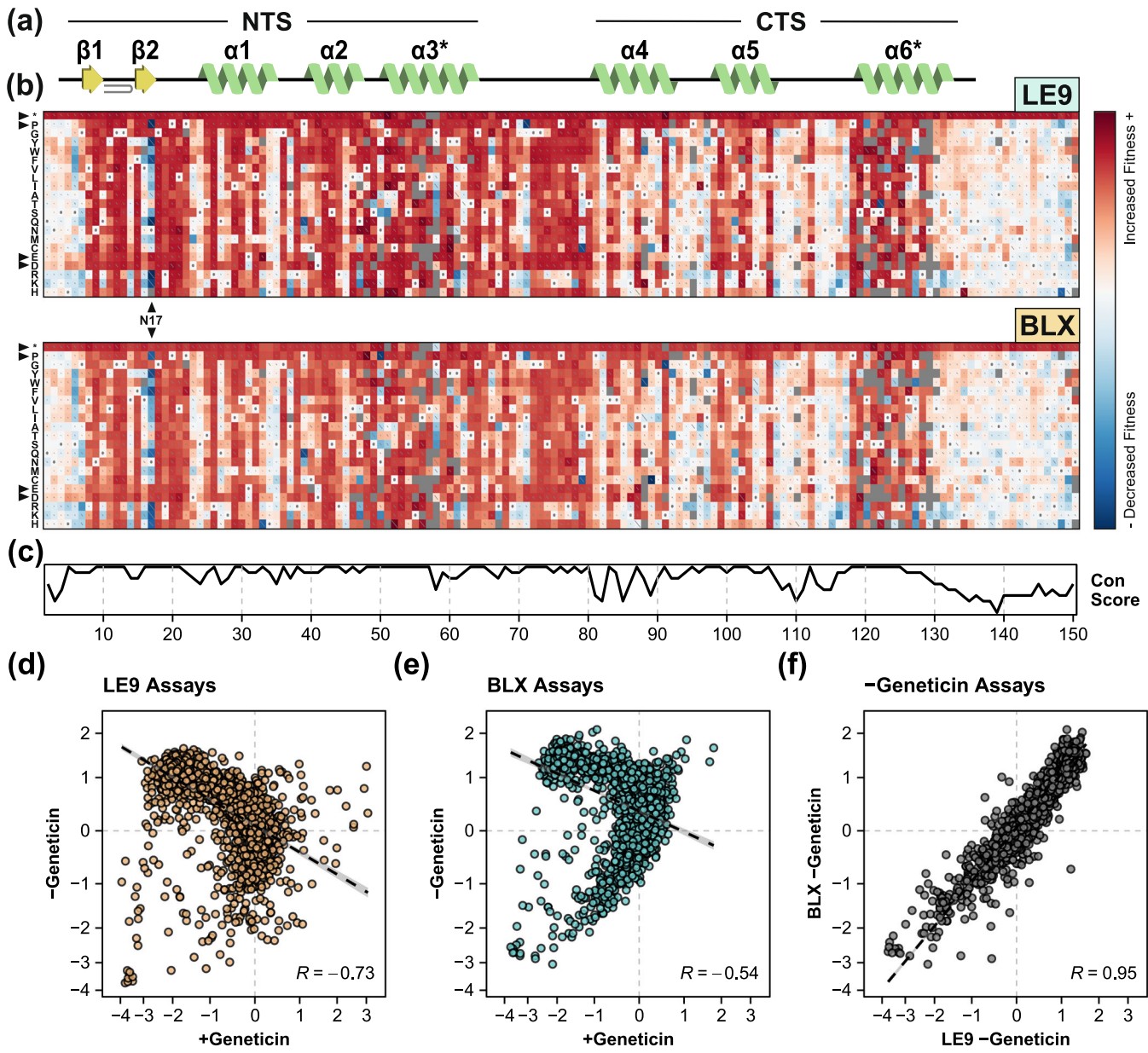

**Figure 4. PAX6 DMS selection pressure without antibiotic.**

(A) Secondary structure of PAX6 paired domain, as in Fig. 2. (B) Sequence-function maps generated following the deep mutational scan of PAX6 in the LE9 strain (top) and BLX strain (bottom) in the absence of geneticin, with the position in PAX6 and amino acid variant plotted on the X and Y axes, respectively. Colours and other symbols as in Fig. 2. (C) PAX6 evolutionary sequence conservation. (D, E) Correlations of variant fitness scores with (+geneticin) and without antibiotic (-geneticin) in the LE9 (D) and BLX (E) assays. (F) Correlation of variant fitness scores between LE9 and BLX assays performed in the absence of geneticin.

remaining 411 variants provide evidence for neither pathogenicity nor benignity.

Following the observation that fitness scores offered accurate classification of pathogenic variants, we explored whether atypical (non-aniridia) phenotypes could be explained using our approach (Fig. 5F,G). Taking advantage of a recent publication in which ocular disease phenotypes were systematically annotated in individuals possessing PAX6 missense variants (Williamson et al, 2019), we subdivided variants into three groups; (1) individuals with mild or classical aniridia; (2) individuals with atypical

phenotypes that included anterior segmentation, MAC (microphthalmia, anophthalmia, coloboma) spectrum, or classical aniridia with any of the following: optic disc coloboma, morning glory disc anomaly, coloboma (iris/choroid/retina), microphthalmia, anophthalmia, or microcornea; (3) individuals where assignation to (1) or (2) could not be determined. No significant difference was detectable in the BLX_ABS assay across variants with mild/classical aniridia or atypical phenotype-linked variants (Fig. 5F). However, atypical phenotype-linked variants had significantly higher ($p = 1.8 \times 10^{-3}$, Wilcoxon test with BH correction) absolute effect

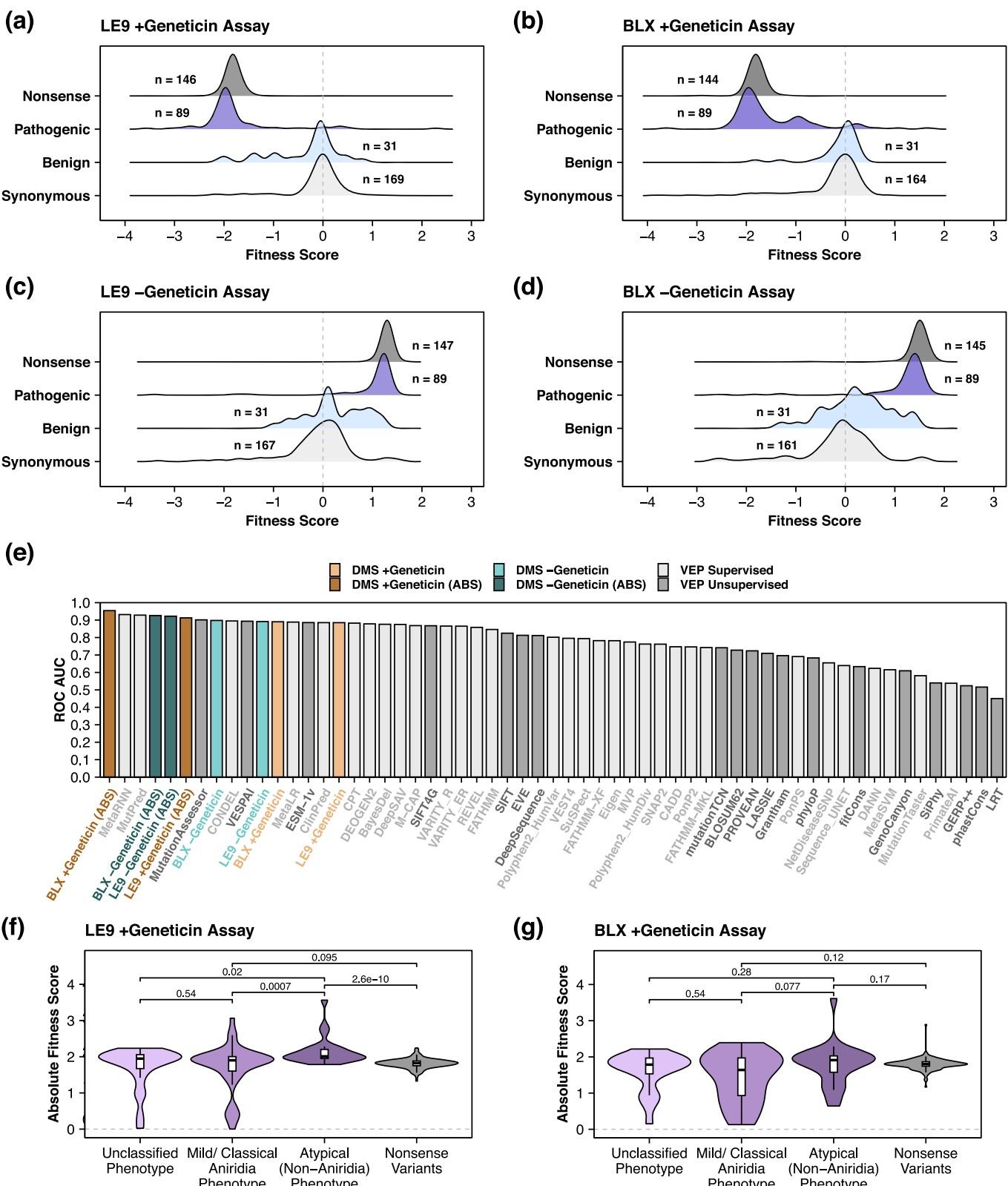

**Figure 5. Classification of benign and pathogenic variants of PAX6 using DMS.**

(A–D) Human PAX6 variants (pathogenic: HGUeye, ClinVar, and HGMD; benign: gnomAD; see Dataset EV1) and their corresponding fitness scores were compared for the indicated assays, using synonymous and nonsense variants to indicate neutral and loss of function, respectively. Dashed lines represent the fitness score for wild-type PAX6. (E) Comparison of DMS assay data ability to accurately classify pathogenic and benign PAX6 variants against supervised and unsupervised variant effect predictors (VEPs), ranked by area under the receiver operating characteristic curve (ROC AUC). $+/-$ geneticin indicates the presence/absence of geneticin in each DMS assay, respectively. "ABS" suffix denotes DMS datasets for which absolute variant fitness scores have been calculated and used in variant classification. (F, G) Absolute fitness scores for pathogenic PAX6 variants subdivided by the corresponding disease phenotype where accurate classifications could be made (see Methods) for $n = 235$ variants in the LE9 (F) and $n = 233$ variants in the BLX (G) assays. Boxes are defined as in Fig. 3. All $p$-values are calculated using the Wilcoxon test and adjusted using FDR. Source data are available online for this figure.

sizes than mild/classical aniridia phenotype-linked variants according to the LE9_ABS assay (Fig. 5G) implying these variants experienced a greater degree of DNA-binding disruption. Additionally, mild/classical aniridia phenotype-linked variants were also indistinguishable from nonsense variants in the LE9_ABS assay while atypical phenotype-linked variants had the greatest absolute change in DNA binding compared to nonsense variants ($p = 1.3 \times 10^{-9}$, Wilcoxon test with BH correction). Taken together, these analyses show a small but statistically significant difference in fitness scores between groups of variants with different clinical phenotypes.

## Discussion

### PAX6 genotype-phenotype relationship

Here we systematically measured the effects of missense mutations in the paired domain of PAX6 on interactions with DNA and we predicted the pathogenicity of known PAX6 variants with 92% accuracy. Of the variants previously observed in the human population, 31 were known as benign and 91 as pathogenic. We quantified the effects of 2574 additional variants and we provide supporting to moderate evidence that 1306 are likely benign and 977 are likely pathogenic.

While heterozygous nonsense mutations in PAX6 typically lead to aniridia, missense mutations cause diverse phenotypes that include microphthalmia, anophthalmia, and coloboma (MAC), isolated foveal hypoplasia, and congenital nystagmus. The reasons for these phenotypic differences are not well understood: in theory, missense mutations can cause partial loss of function, gain-of-function, or dominant negative effects, and indeed functional studies show that some missense mutations lead to different DNA binding outcomes (Kejun Tang et al, 1997; Williamson et al, 2019). In the present study, approximately 20% of missense mutations caused partial loss of function and an additional 9–16% showed either increased binding to DNA or toxicity, which could both be considered as molecular gain-of-function. There is also indication that specific phenotypes in yeast can be related to patients' phenotypes. For example, variants N17S and N17K were found respectively in aniridia and Peter's anomaly; in our data, N17S caused a LOF or near-LOF phenotype, consistent with aniridia, whereas N17K was toxic to both LE9 and BLX strains, consistent with its more severe patient phenotype. Our results are also consistent with the survival of a patient with compound heterozygous mutations in PAX6 (R38W/R240*), since the R38W mutation showed some activity in the BLX strain (Lima Cunha et al, 2021). Overall, statistical analysis of the yeast data showed a

moderately strong association between atypical patient phenotypes and fitness scores in the LE9, but not BLX strains.

PAX6 exists in two major splicing isoforms, canonical PAX6 and PAX6(5a), which differ in their expression patterns: the canonical isoform is expressed during embryonic development and controls differentiation and cell fate determination, whereas PAX6(5a) is expressed at later stages and plays a role in cell proliferation. It has been reported that the two isoforms also differ in their mode of binding to DNA, with the canonical isoform preferentially binding through the NTS and PAX6(5a) through the CTS domain. In our experiments with canonical PAX6, we indeed observed a significantly higher effect of mutations in the NTS than in CTS, in both the LE9 and BLX strains. However, many CTS mutations, especially in the α6 helix, also disrupted binding, showing that both NTS and CTS domains contribute to canonical PAX6:DNA interactions.

### PAX6 mutations alter DNA binding specificity

A possible explanation for the phenotypic diversity of missense mutations is that they may change DNA binding specificity of PAX6, leading to stronger binding of certain enhancer elements and weaker binding of others, which may in turn lead to variant-specific changes in development. A similar phenomenon has been reported for a missense variant in the transcription factor Krüppel-like factor 1 that is important in erythroid cell development: the variant caused an autosomal dominant disorder by corruption of binding preference from a canonical DNA motif, leading to dysregulation of gene expression (Ilsley et al, 2019). In PAX6, Tang et al, found that the I87R variant found in a patient with aniridia ablated DNA-binding, while R26G, a variant located proximally to the protein-DNA interface that caused relatively less disruption DNA-binding, lead to a non-aniridia phenotype(Kejun Tang et al, 1997). Similarly, a recent study demonstrated that the MAC-associated variant, S54R, showed sequence-dependent DNA-binding disruption to two characterised PAX6-binding sites, LE9 and SIMO, while aniridia-associated variants such as C52R and S121L consistently ablated DNA-binding (Williamson et al, 2019).

In our study, most variants that perturbed DNA binding did so in all four conditions: in LE9 and BLX strains, with and without geneticin. This suggests that single mutations are more likely to influence the strength of DNA binding in general, rather than influence sequence specificity. However, we also found numerous examples of mutations that either alter PAX6 interactions with one of the DNA response elements without changing interactions with the other element, or that affected promiscuous binding to the yeast genome without affecting the binding to LE9 or BLX elements. In agreement with the findings by Williamson et al (2019), we

observed that the mutation S54R results in a substantial loss of affinity to LE9, however, rather than an inconsequential effect on binding on another element observed previously, we saw increased binding to BLX. The most striking position in which we saw PAX6 target-specific effects was at I71, where numerous amino acid substitutions resulted in substantially increased affinity towards LE9, but reduced affinity to BLX. No human variants to date have been identified at this position, however, the nearby pathogenic variant S74G showed sequence-specific inverse effects on DNA-binding in our assay. In patients, S74G is associated with non-aniridia phenotypes, including ocular, neurological and cognitive impairments (Dansault et al, 2007). Interestingly, a recent study exploring the link between DNA-binding affinity and phenotypic differences found there was no dramatic difference in the affinity of wild-type PAX6 and S74G to three DNA targets, however, these targets did not include LE9 or BLX (Lee et al, 2020).

The variant G51R has previously been reported in individuals with MAC (Williamson et al, 2019), and according to our assay, results in a LOF in the LE9 strain and a neutral effect in BLX which might explain the atypical phenotype under the hypothesis that DNA-binding exclusively causes pleiotropy. However, G51R has also been reported in individuals with aniridia (Chen and Zhu, 2016) suggesting other factors affecting phenotypic heterogeneity. Intrafamilial phenotypic variability associated with the same mutation has been reported both in individuals with aniridia harbouring indel mutations as well as in individuals with missense variants that cause atypical phenotypes such as isolated foveal hypoplasia and nystagmus (Lima Cunha et al, 2021; Pedersen et al, 2019, 2018). Mechanisms such as mosaicism, gene dosage and genetic modifiers are possible explanations of variability (Lima Cunha et al, 2021; Tarilonte et al, 2018; Chou et al, 2015) so it is conceivable that pathogenic variants such as G51R are subject to similar confounding layers of control.

## Effect of PAX6 in the absence of antibiotic

Our assay was designed to probe PAX6:DNA interactions through an antibiotic resistance reporter gene, but it also showed effects of PAX6 mutations in the absence of antibiotic. A similar phenomenon was described in *Escherichia coli*, where expression of variants of *TEM-1* β-lactamase antibiotic resistance gene led to collateral fitness effects in the absence of antibiotic (Mehlhoff et al, 2020). Whereas missense mutations in *TEM-1* reduced growth through various mechanisms, we speculate that PAX6 reduces growth by promiscuous binding to yeast genomic DNA and perturbing the transcriptome. Overexpression of GAL4 in yeast was previously observed to cause a growth defect (Gill and Ptashne, 1988), but the effect, known as squelching, was independent of GAL4:DNA binding and instead resulted from titration of transcription factors by the activating region of GAL4. In our study, expression of DNA binding-deficient PAX6-GAL4 fusions did not reduce growth, suggesting that the effect we observe is distinct from squelching.

PAX6 mutant phenotypes with and without antibiotic were negatively correlated with each other, but the correlation was strongly nonlinear. In the BLX strain, variants with an intermediate level of fitness without antibiotic appeared to have the highest fitness with antibiotic; a similar but less pronounced pattern was seen in the LE9 strain. In both strains, many variants behaved like wild-type in the presence of antibiotic, but had either negative or positive effects in the absence of antibiotic. In addition, some variants negatively affected growth both in the presence and absence of antibiotic. Possibly, these variants do not bind to the bait sequence, but retain promiscuous binding to the yeast genome. While we don't clearly understand the causes of these nonlinear correlations, the pattern is broadly reproducible between the LE9 and BLX strains and it indicates the complex interplay between the growth promoting effects of PAX6 through expression of the antibiotic resistance gene, and growth-reducing effects caused by promiscuous binding to yeast genomic DNA.

Regardless of the mechanism, these results suggest that yeast growth in the absence of antibiotic selection can be a useful phenotypic readout for deep mutational scanning of other human transcription factors. Unlike the normal Y1H setup, which necessitates laborious design and testing of strains with appropriate bait sequences, the assay without antibiotic is predicted to work without a bait sequence and could potentially be applied directly to many different TFs. Of the 1600 human TFs, 313 have been associated with at least one disease phenotype according to a recent study (Lambert et al, 2018), and therefore may be good targets for this style of assay.

## Uses of deep mutational scanning in an age of variant effect predictors

Large-scale genome sequencing has greatly increased the number of genetic variants detected in the human population but it also caused a proliferation of variants of uncertain significance (VUS). DMS and VEPs are two approaches that have the potential to resolve VUSs at scale by classifying them as either benign or pathogenic, but classification accuracy depends on the target gene and on the type of VEP or DMS approach being used. For some genes such as BRCA1, DMS performs better than all computational predictors (Findlay et al, 2018), whereas computational methods outperform DMS for other genes (Livesey and Marsh, 2020). This raises questions about what experimental or computational variables influence the performance of DMS assays, and about the utility of DMS in general, given that as an experimental approach DMS is much more resource-intensive than computational predictors.

The four DMS datasets we collected showed consistently high performance in their ability to predict pathogenic variants in PAX6, with ROC AUC values between 0.885 and 0.954. This performance was slightly better than that of the top unsupervised predictors identified in a recent benchmarking study (EVE, ESM-1v and DeepSequence, ROC AUC 0.811–0.886) (Livesey and Marsh, 2023; Frazer et al, 2021; Meier et al, 2021; Riesselman et al, 2018), but were on average slightly worse than top supervised predictors (MetaRNN, MutPred, and MutationAssessor, ROC AUC 0.901–0.932) (Li et al, 2022; Pejaver et al, 2020; Reva et al, 2011). These top supervised predictors may, however, have been trained on data used for evaluation and suffer from data circularity (Livesey and Marsh, 2022b). We identified several factors that influenced the predictive performance of DMS: (1) higher data coverage (sequencing depth and more importantly, number of barcodes) for a particular variant increased predictive power, though applying a coverage threshold also decreased the proportion of variants we could classify; (2) the number of experimental time points used for fitness calculations had a complex relationship with performance:

too few data points resulted in lower accuracy, but too many data points reduced the numbers of fitness scores obtained, seemingly due to difficulty the model had in fitting all the data; (3) as found in some previous studies (Livesey and Marsh, 2022a; Gerasimavicius et al, 2020), the use of absolute fitness scores instead of raw scores substantially increased the predictive performance (ROC AUC for raw scores, 0.885–0.891; absolute scores, 0.913–0.954). We propose that the utility of absolute scoring depends on whether the gene in question harbours pathogenic gain-of-function (GOF) variants: transcription factors are known to contain such mutations, and indeed examples of pathogenic gain-of-function variants have also been described in PAX6 (Williamson et al, 2019). Owing to the dosage sensitivity of PAX6, it has been shown that both overexpression (Manuel et al, 2008; Davis and Piatigorsky, 2011; Schedl et al, 1996) and heterozygous knockouts (Jones et al, 2002; Chauhan et al, 2002) can be pathogenic, so it is conceivable that any deviations in missense variant activity relative to homozygous wild-type functional levels, positive or negative, has the potential to be pathogenic.

Irrespective of its ability to predict disease variants, one advantage of DMS over the current generation of VEPs is that DMS can probe multiple molecular and cellular phenotypes in parallel. This can lead to a better mechanistic understanding of genetic variants, e.g. their effects on cell growth and survival, gene expression, protein-protein interactions and protein-DNA interactions (Findlay et al, 2018; Li et al, 2019; Faure et al, 2022). DMS can also probe the effects of mutations depending on the cellular context, drug treatment, or the presence of additional mutations (Zhang et al, 2020; Puchta et al, 2016). While we have not investigated double mutants systematically in this study, our dataset contains thousands of such mutants which could be analysed for intragenic epistasis (Puchta et al, 2016) in the DNA-binding domain of PAX6.

DMS also has limitations: for example, any mutations that influence splicing of human PAX6 would not have been detected in the present assay that used an intronless PAX6 cDNA in a yeast model system. Effects on splicing are detectable using other DMS techniques, such as saturation genome editing (Findlay et al, 2018), and might be detected by a subset of generic VEPs or dedicated splicing prediction software (Desmet et al, 2009; Cheng et al, 2019). In addition, we would not have detected effects of mutations on interactions of PAX6 with cofactors (such as SOX2) that are not found in yeast. An effect on splicing or cofactor interaction could explain the only known pathogenic variant of PAX6 (E93K) that was scored as benign in our analysis. Nevertheless, the high predictive power of our assays demonstrates that most pathogenic mutations in the paired domain of PAX6 can be explained simply by the effects of these mutations on PAX6:DNA association.

# Methods

## Enhancer-specific *S. cerevisiae* strain generation

All strains were derived from the S288C strain, Y1HGold (MATα, ura3-52, his3-200, ade2-101, trp1-901, leu2-3, 112, gal4Δ, gal80Δ, met–, MEL1), obtained from Takara Bio. Initially the Matchmaker® Gold Yeast One-Hybrid Library Screening System (Takara Bio) was

employed, however, owing to the incompatibility of Aureobasidin A with liquid cultures, an equivalent geneticin resistance gene was subcloned in place of the Aureobasidin A resistance gene. PAX6 DNA response elements (baits) were synthesised either as single copies or tandem triple repeats (IDT) and individually cloned into a plasmid containing a downstream geneticin resistance gene and URA3 marker (pKani). Each pKani+bait vector was then integrated into Y1HGold genomic DNA via homologous recombination. Each pKani+bait vector was linearised by restriction digestion with BstBI and incubation at 65 °C for 1 h, followed by purification using DNA Clean & Concentrator-5 (Zymo). DNA was then transformed into Y1HGold using the protocol described by Gietz and Schiestl (2007), followed by selection on plates lacking uracil. Correct genomic integration was confirmed using colony PCR.

## PAX6 variant library construction

The PAX6 (1–150) variant library was generated using a modified version of PCR-based saturation mutagenesis described by Wrenbeck et al (2016). Briefly, the top strand of a dsDNA plasmid (pAINt1) containing wild-type PAX6 was nicked using Nt.BbvCI, and degraded using Exonuclease I and III (NEB). The resulting ssDNA template was then split between 6 parallel reactions and custom mutagenic oligo primers (IDT) added such that they spanned 6 adjacent 75 bp stretches of the entire PAX6 sequence construct (Appendix Table S1). Phusion polymerase (NEB) was then used to synthesise a mutated top strand, followed by Taq DNA ligase (NEB) to seal nicks. The 6 reactions were then pooled and the wild-type bottom strand nicked using Nb.BbvCI and digested using Exonuclease I and III. A second primer was then added and a complementary mutated bottom strand synthesised using Phusion High-Fidelity PCR Master Mix with HF Buffer (NEB) and Taq DNA ligase, creating a pooled dsDNA PAX6 variant library. This library was then transformed into XL10-Gold Ultracompetent Cells (Agilent) and transformants grown on appropriate antibiotic plates. Colonies were harvested by resuspension in $H_2O$, followed by scraping, pooling, and centrifugation. Finally, plasmids were isolated using Qiaprep spin miniprep kit (Qiagen).

## Assigning barcodes to variants

The PAX6 variant library within pAINt (Dataset EV2) was transferred to a centromeric yeast expression plasmid (pMP1_XL) (Dataset EV3) containing approximately $1.2 \times 10^7$ unique 30-nt barcodes, using the LR Gateway system (Thermo Scientific). The reaction product was then transformed into Library Efficiency™ DH5α Competent Cells (Thermo Scientific) and plated on appropriate antibiotic plates. Approximately $7.5 \times 10^4$ colonies were harvested by resuspension in H2O, followed by scraping, pooling, and centrifugation.

An aliquot of the barcoded PAX6 variant library was taken and a region of the plasmid containing the barcodes, PAX6, and GAL4 was excised from the vector using restriction digestions followed by BluePippin purification. This fragment was then sequenced using PacBio Sequel II and 20 h of movie collection time. Finally, barcode:variant phasing was achieved using the alignparse Python package (Crawford and Bloom, 2019).

## Competition assays

Competition assays were performed using a complete supplement mixture without histidine (Formedium) and yeast nitrogen base without amino acids (Formedium) media, supplemented with 2% glucose. To begin, the barcoded PAX6 variant library was transformed into two separate strains, LE9 and BLX, in two independent assays. The transformation process followed the protocol described in the Yeastmaker™ Yeast Transformation System 2 User Manual (Takara Bio), yielding approximately $2 \times 10^6$ and $7 \times 10^6$ transformants for LE9 and BLX strains, respectively. Transformed cells were then inoculated into 500 ml of liquid media and incubated at 30 °C and 230 RPM for 12–24 h until reaching an OD600 of approximately 1.0. Subsequently, approximately $7.5 \times 10^8$ cells were transferred to fresh media and grown for an additional 12 h (T0). The T0 cultures were then split into two parallel lineages. All cultures were then maintained between an OD600 of 0.05 and 1.0 by passaging every 12 h for a total of 36 h (T1-3). Geneticin was then added to one lineage of each strain (LE9; 400 μg/ml, BLX; 500 μg/ml) starting from T0. Just before each passage, 10 to 20 OD600 units of cells were collected and pelleted by centrifugation at $700 \times g$ for 5 min and the supernatant removed, followed by two rounds washing using resuspension in $H_2O$ and centrifugation to remove extracellular plasmid DNA.

## Plasmid isolation and sequencing sample preparation

Plasmid isolation was achieved using the protocol described by Fowler et al (2014), which in summary used a freeze-thaw cycle and zymolyase cell wall digestion steps coupled with Qiaprep spin miniprep kit (Qiagen) column purification. Samples from each timepoint were prepared for next-generation sequencing using a single limited-cycle PCR. Barcodes were amplified using primers possessing overhangs that contained sequencing primer annealing sequences, unique dual indexes, and P5/P7 Illumina adapter sequences (Appendix Table S1). 50 μl PCR reactions were performed for each timepoint using Phusion High-Fidelity PCR Master Mix with HF Buffer (NEB) with 20 to 50 ng of template isolated plasmid DNA. Each reaction was then thermocycled using the following conditions: (1) initial denaturation at 98 °C for 30 s, (2) incubation at 98 °C for 8 s, (3) incubation at 69 °C for 20 s, (4) incubation at 72 °C for 8 s, (5) incubation at 72 °C for 7 min. Steps 2 to 4 were repeated for 12 to 19 cycles (<20). Multiple replicate 50 μl reactions were performed for each timepoint to ensure enough PCR product was generated. 2.5 μl of Exo I nuclease (NEB) was then added directly to each 'dirty' PCR product and incubated at 37 °C for 1 h, followed by purification using QIAquick PCR Purification Kit (Qiagen). Samples were further purified by 2% E-Gel SizeSelect agarose gel (Invitrogen). The resulting products were purified using QIAquick PCR Purification Kit (Qiagen), after which all samples were pooled at equimolar amounts (quantified by QuBit [Invitrogen]) and then quantitated by Bioanalyzer (Agilent).

## Barcode sequencing and fitness score calculations

Samples were sequenced using either Nextseq 550 High Output flowcell with 75 cycles or Nextseq 2000 P2 flowcell with 100 cycles for the LE9 and BLX assays, respectively. A custom Read 1 sequencing primer (Appendix Table S1) was used in both libraries, and annealed to the template such that the first position in the read corresponded to the first nucleotide of the barcode. Samples were spiked with 10% PhiX. Following demultiplexing, the regions flanking the barcodes were trimmed using trimgalore (Krueger et al, 2023) then barcodes counted and variant fitness scores calculated using the Enrich2 tool (Rubin et al, 2017). The first four time points of each experiment were used ($T = 0$, 12, 24 and 36 h) and a minimum threshold of 25 reads per barcode was applied to the first timepoint ($T = 0$). $T = 0$ represents the time when antibiotic was added to the media and the first sample was collected.

In addition to the *p*-values generated by Enrich2 for each variant using z-distribution under the null hypothesis that it behaves like wild-type ($p_{WT}$), an equivalent *p*-value was calculated under the null hypothesis that each variant behaves like a nonsense variant ($p_{nonsense}$). All *p*-values were adjusted to *q*-values using the fdr tool from the p.adjust() r package (Benjamini and Hochberg, 1995). Relative changes in function for each variant were assigned for each assay using the following categories: increased (fitness score > 0, & $q_{WT} < 0.05$), neutral ($q_{WT} > 0.05$), hypomorphic (0 > fitness score > mean nonsense score, & $q_{WT} < 0.05$, & $q_{nonsense} < 0.05$), LOF ($q_{nonsense} > 0.05$), and toxic (fitness score < mean nonsense score, & $q_{nonsense} < 0.05$).

## Structural analysis of fitness scores

PAX6 region amino acid residue ranges were defined as: NTS (4–63), Linker (64–79), CTS (80–136) and 'other' (1–3 and 137–150) according to the 6pax crystal structure (Xu et al, 1999). LE9 and BLX DNA sequences were modelled with PAX6 using the same structure. A solvent accessibility surface area threshold of >25% was used to classify variants as solvent accessible and <25% as solvent inaccessible. Distances between PAX6 residues and DNA were calculated by measuring between each amino acid residue and the nearest non-hydrogen atom of the DNA duplex. A median fitness score across each position was calculated, mapped to either the PAX6:LE9 or PAX6:BLX structure, and visualised with PyMol (The PyMOL Molecular Graphics System, Version 2.0 Schrödinger, LLC).

## Human variants and phenotypes

Human pathogenic PAX6 variants and their respective annotated phenotypes were extracted from a previous publication (Williamson et al, 2019) using the HGUeye cohort ($n = 67$), and aggregated with more recently identified variants in ClinVar ($n = 17$) and the Human Gene Mutation Database (HGMD) ($n = 7$) (Dataset EV1). Variants were subdivided into three groups that corresponded to different phenotypes based on the above publication (Williamson et al, 2019). Group one comprised variants associated with mild or classical aniridia, while group two comprised of individuals with non-aniridia (atypical) phenotypes that included anterior segmentation and MAC (microphthalmia, anophthalmia, coloboma) spectrum, or classical aniridia with any of the following: optic disc coloboma, morning glory disc anomaly, coloboma (iris/choroid/retina), microphthalmia, anophthalmia, or microcornea. The third group comprised variants corresponding to individuals where categorisation into group one or two could not be determined. Benign variants were taken from the gnomAD database ($n = 32$). See also Dataset EV1 for a list of all gnomAD and their annotations.

## VEP benchmarking

VEPs used for benchmarking are listed in Appendix Table S3. Most VEP scores were obtained from the dbNSFP database version 4.2 (Liu et al, 2020). To generate amino acid level scores from the database, the score for each VEP across all codons resulting in the same amino acid were averaged. DeepSequence, SIFT, ESM-1v and sequence_unet were run locally. Publicly available pre-computed scores were used for VESPAl, CPT, DeepSAV, EVE, PonP2, mutationTCN and LASSIE. Eigen is classified as supervised VEP due to its inclusion of PolyPhen-2 as a predictive feature. To calculate the ROC AUC for DMS assays and VEPs, we used the roc_auc_score function of the Python package sklearn, with pathogenic variants labelled as class '1' and benign variants as class '0'. Due to variability in predictor coverage of PAX6, a minimum of 10 pathogenic and 10 benign variant predictions were required for the inclusion of each VEP. Predictor scale orientations were normalised by inversion where appropriate to maintain comparability, by application of the following formula:

$$\text{inverted predictions} = \min(\text{predictions}) - \text{predictions} + \max(\text{predictions})$$

The optimal classification threshold was calculated from the ROC curve by identifying the threshold that generated the point closest to the top left of the curve (the highest true positive rate − false positive rate). Correctly classified pathogenic variants produced scores greater than, or equal to these threshold values, while correctly classified benign variants scored less than the threshold. Accuracy was also calculated at this threshold using the following formula:

$$\text{accuracy} = \frac{(\text{true positives} + \text{true negative})}{\text{total classified variants}}$$

## Estimation of the evidence of pathogenicity and benignity

All variants with a pathogenic or benign label were ordered by descending fitness score using the absolute scores from the BLX assay with geneticin. The prior probability of pathogenicity was calculated from this data as the proportion of pathogenic variants. Windows of dynamic size were then constructed around each variant in the labelled dataset using a similar method to Pejaver et al (2022). Briefly, windows were required to contain at least 20 variants, and at least two from the minority class. If these conditions were not met then the window size was increased to encompass more variants until they were achieved. The posterior probability of pathogenicity was calculated within each window as the proportion of pathogenic variants. As per the method described by Pejaver et al (2022), the number of benign variants were weighted to calibrate for the prior probability of pathogenicity using the formula:

$$w = \frac{(1 - P1) * n_{\text{pathogenic}}}{n_{\text{benign}} * P1}$$

Where $P1$ is the prior probability of pathogenicity, $n_{\text{pathogenic}}$ is the total number of pathogenic variants and $n_{\text{benign}}$ is the total number of benign variants.

The odds of pathogenicity (oddsPath) was then calculated within each window using the formula:

$$\text{oddsPath} = \frac{P2 * (1 - P1)}{(1 - P2) * P1}$$

Where $P1$ is the prior probability of pathogenicity and $P2$ is the posterior probability of pathogenicity.

We then applied the resulting oddsPath results to thresholds in a recent recommendation for an update to the American College of Medical Genetics and Genomic (ACMG)/Association for Molecular Pathology (AMP) guidelines (Richards et al, 2015; Brnich et al, 2019) to obtain score thresholds for different evidence strength boundaries. These thresholds were then applied to the full dataset to estimate the evidence strength. The process was then repeated for benign variants with the prior probability of benignity, posterior probability of benignity and windows being determined from the lowest DMS score rather than the highest. Thresholds for pathogenic evidence were calculated from the pathogenic data while thresholds for the benign evidence were calculated from the benign data. Data that did not meet the threshold of 'supporting' for either pathogenic or benign was labelled as indeterminate.

## Sequence alignment

The evolutionary sequence conservation of PAX6 was aligned with ConSurf (Ashkenazy et al, 2016) using MAFFT and homologues collected from UNIREF90 and HMMER search algorithm.

## Data availability

The datasets and computer code produced in this study are available in the following databases: Raw barcode sequencing files, barcode counts, and barcode-variant maps: Gene Expression Ominbus GSE253580. Variant scores and annotations for four experiments (LE9 and BLX strains, with and without geneticin): MaveDB (Esposito et al, 2019) urn:mavedb:00000665-a (https://www.mavedb.org/#/experiments/urn:mavedb:00000665-a).

The source data of this paper are collected in the following database record: biostudies:S-SCDT-10_1038-S44320-024-00043-8.

## Peer review information

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

## Acknowledgements

We are grateful to the IGC Sequencing facility, the Edinburgh Genomics Facility, and the DNA Technologies and Expression Analysis Core at UC Davis Genome Center for their expert support. We acknowledge the following funding sources: UK Medical Research Council (MRC) University Unit programmes MC_UU_00035/8 (GK) and MC_UU_00035/9 (JAM), Wellcome Trust (fellowship 207507 to GK), European Research Council (ERC) under the European Union's Horizon 2020 research and innovation programme (grant agreement No. 101001169 to JAM). AFMD was supported by the Precision Medicine PhD programme at the University of Edinburgh.

## Author contributions

**Alexander F McDonnell**: Conceptualization; Formal analysis; Investigation; Visualization; Methodology; Writing—review and editing. **Marcin Plech**: Investigation; Methodology. **Benjamin J Livesey**: Data curation; Formal analysis; Writing—review and editing. **Lukas Gerasimavicius**: Formal analysis. **Liusaidh J Owen**: Resources. **Hildegard Nikki Hall**: Resources. **David R FitzPatrick**: Conceptualization; Supervision; Methodology. **Joseph A Marsh**: Supervision; Methodology; Writing—review and editing. **Grzegorz Kudla**: Conceptualization; Supervision; Funding acquisition; Investigation; Methodology; Writing—original draft; Writing—review and editing.

Source data underlying figure panels in this paper may have individual authorship assigned. Where available, figure panel/source data authorship is listed in the following database record: biostudies:S-SCDT-10_1038-S44320-024-00043-8.

## Disclosure and competing interests statement

The authors declare no competing interests.

# Expanded View Figures

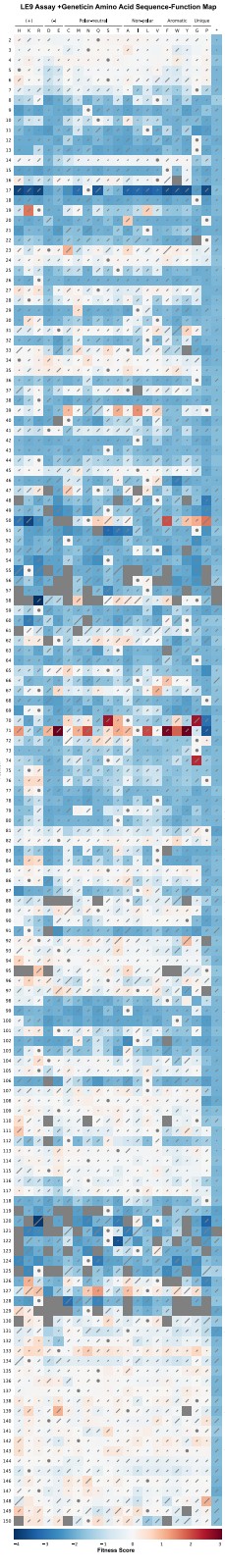

**Figure EV1. Variant effect map of PAX6 paired domain in LE9 strain with geneticin treatment.**

Colours and symbols as in Fig. 2. Source data are available online for this figure.

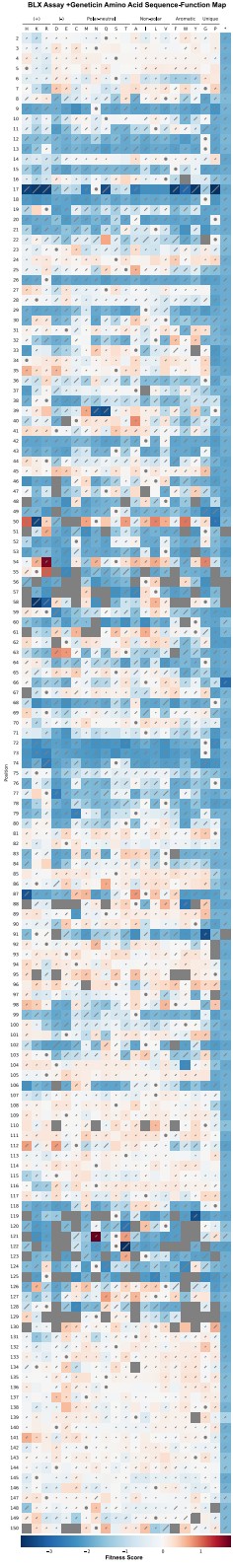

**Figure EV2. Variant effect map of PAX6 paired domain in BLX strain with geneticin treatment.**

Colours and symbols as in Fig. 2.

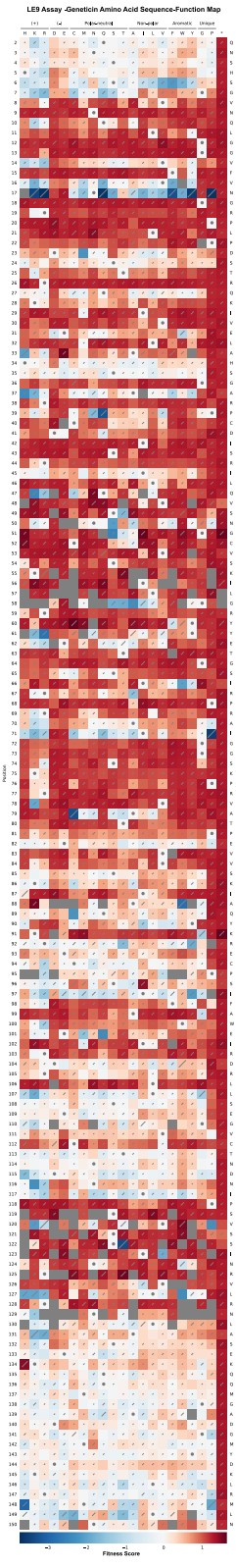

**Figure EV3. Variant effect map of PAX6 paired domain in LE9 strain without geneticin.**

Colours and symbols as in Fig. 2.

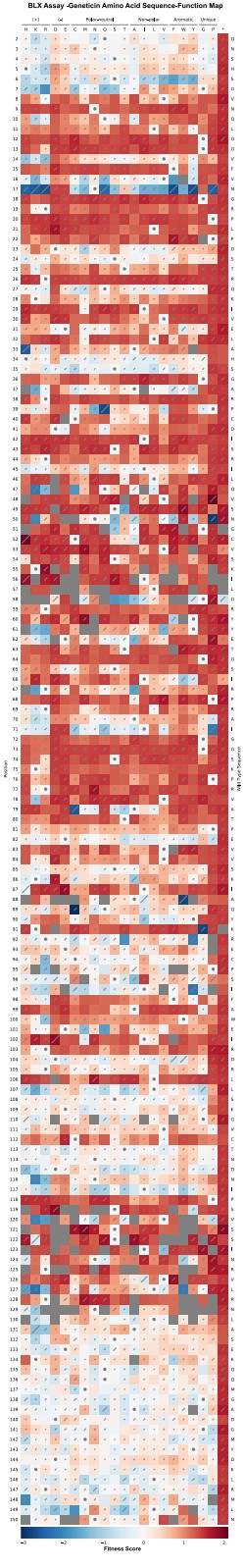

**Figure EV4. Variant effect map of PAX6 paired domain in BLX strain without geneticin.**

Colours and symbols as in Fig. 2.

