## [Peer Review File · Molecular Systems Biology]

Deep mutational scanning quantifies DNA binding and predicts clinical outcomes of PAX6 variants

Alexander McDonnell, Marcin Plech, Benjamin Livesey, Lukas Gerasimavicius, Liusaidh Owen, Hildegard Hall, David FitzPatrick, Joseph Marsh, and Grzegorz Kudla

Corresponding author(s): Grzegorz Kudla (gkudla@gmail.com)

Review Timeline:

Submission Date:	4th Sep 23
Editorial Decision:	12th Oct 23
Revision Received:	25th Jan 24
Editorial Decision:	1st Mar 24
Revision Received:	5th Apr 24
Accepted:	14th May 24

Editor: Poonam Bheda

Transaction Report:

12th Oct 2023

Dear Dr. Kudla,

Thank you again for submitting your work to Molecular Systems Biology. We have now heard back from the three reviewers who agreed to evaluate your study. As you will see below, the reviewers appreciate that the proposed approach addresses a timely topic. However, they raise a series of concerns, which we would ask you to address in a major revision.

I think that the recommendations of the reviewers are rather clear and I therefore do not see the need to repeat the comments listed below. In line with Reviewer 3, please ensure that the sequencing data and variant scores are available in an acceptable repository.

We require:

4) A .docx formatted letter INCLUDING the reviewers' reports and your detailed point-by-point responses to their comments. As part of the EMBO Press transparent editorial process, the point-by-point response is part of the Review Process File (RPF), which will be published alongside your paper.

5) A complete author checklist, which you can download from our author guidelines (<https://www.embopress.org/page/journal/17574684/authorguide#submissionofrevisions>). Please insert information in the checklist that is also reflected in the manuscript. The completed author checklist will also be part of the RPF.

6) Please note that all corresponding authors are required to supply an ORCID ID for their name upon submission of a revised manuscript.

7) It is mandatory to include a 'Data Availability' section after the Materials and Methods. Before submitting your revision, primary datasets produced in this study need to be deposited in an appropriate public database, and the accession numbers and database listed under 'Data Availability'. Please remember to provide a reviewer password if the datasets are not yet public (see <https://www.embopress.org/page/journal/17574684/authorguide#dataavailability>).

In case you have no data that requires deposition in a public database, please state so in this section. Note that the Data Availability Section is restricted to new primary data that are part of this study. This study includes no data deposited in external repositories.

8) For data quantification: please specify the name of the statistical test used to generate error bars and P values, the number (n) of independent experiments (specify technical or biological replicates) underlying each data point and the test used to calculate p-values in each figure legend. The figure legends should contain a basic description of n, P and the test applied. Graphs must include a description of the bars and the error bars (s.d., s.e.m.). Please provide exact p values.

10) We replaced Supplementary Information with Expanded View (EV) Figures and Tables that are collapsible/expandable online. A maximum of 5 EV Figures can be typeset. EV Figures should be cited as 'Figure EV1, Figure EV2' etc... in the text and

their respective legends should be included in the main text after the legends of regular figures.

<https://www.embopress.org/page/journal/17574684/authorguide#expandedview>

11) For more information: There is space at the end of each article to list relevant web links for further consultation by our readers. Could you identify some relevant ones and provide such information as well? Some examples are patient associations, relevant databases, OMIM/proteins/genes links, author's websites, etc...

12) Author contributions: CRediT has replaced the traditional author contributions section because it offers a systematic machine readable author contributions format that allows for more effective research assessment. Please remove the Authors Contributions from the manuscript and use the free text boxes beneath each contributing author's name in our system to add specific details on the author's contribution. More information is available in our guide to authors.

13) Disclosure statement and competing interests: We updated our journal's competing interests policy in January 2022 and request authors to consider both actual and perceived competing interests. Please review the policy

<https://www.embopress.org/competing-interests> and update your competing interests if necessary.

14) Every published paper now includes a 'Synopsis' to further enhance discoverability. Synopses are displayed on the journal webpage and are freely accessible to all readers. They include a short stand first (maximum of 300 characters, including space) as well as 2-5 one-sentences bullet points that summarizes the paper. Please write the bullet points to summarize the key NEW findings. They should be designed to be complementary to the abstract - i.e. not repeat the same text. We encourage inclusion of key acronyms and quantitative information (maximum of 30 words / bullet point). Please use the passive voice. Please attach these in a separate file or send them by email, we will incorporate them accordingly.

Share synopsis text and image, as well as eTOC:

Please note that these would be the final versions and changes during proofing are usually not allowed

15) As part of the EMBO Publications transparent editorial process initiative (see our Editorial at

<http://embomolmed.embopress.org/content/2/9/329>), Molecular Systems Biology Medicine will publish online a Review Process File (RPF) to accompany accepted manuscripts.

In the event of acceptance, this file will be published in conjunction with your paper and will include the anonymous referee reports, your point-by-point response and all pertinent correspondence relating to the manuscript. Let us know whether you agree with the publication of the RPF and as here, if you want to remove or not any figures from it prior to publication.

Molecular Systems Biology has a "scooping protection" policy, whereby similar findings that are published by others during review or revision are not a criterion for rejection. Should you decide to submit a revised version, I do ask that you get in touch after three months if you have not completed it, to update us on the status.

I look forward to receiving your revised manuscript.

Yours sincerely,

Poonam Bheda, PhD
Scientific Editor
Molecular Systems Biology

***** PLEASE NOTE ***** As part of the EMBO Press transparent editorial process initiative (see our Editorial at <https://dx.doi.org/10.1038/msb.2010.72>), Molecular Systems Biology publishes online a Review Process File with each accepted manuscripts. This file will be published in conjunction with your paper and will include the anonymous referee reports, your point-by-point response and all pertinent correspondence relating to the manuscript. If you do NOT want this File to be

published, please inform the editorial office at msb@embo.org within 14 days upon receipt of the present letter.

Reviewer #1:

McDonnell et. al present an evaluation of missense variant effects of the PD domain of the PAX6 transcription factor in three unique contexts, including the evaluation of PAX6 binding to two different DNA elements (LE9 and BLX) via yeast-1-hybrid assays, and a fitness assay. First, the barcoded variant library of the PD domain was generated in-house using PCR-based saturation mutagenesis. Next, deep mutational scanning was used to evaluate variants in high-throughput, selectable yeast systems, either by functional variants driving antibiotic resistance, or via fitness. Barcode sequencing and analysis of the allele frequencies over time points generated the sequence-function maps. The results agreed with biochemical expectations, considering that the nonsense and synonymous variants were well-separated in the assays and that deleterious variant scores were found in DNA-proximal positions. Known benign and pathogenic variants of PAX6 performed as expected in the assays. The functional scores of some variant positions differed between the two Y1H assays, which was likely due to differences in DNA-binding specificity. All assays performed better or on par with computational predictors of variant effects, with the best performer being the BLX fitness assay. The sequence-function map was able to provide support for over 2000 missense variants to more clinically-useful classifications such as likely benign or likely pathogenic, according to ACMG guidelines. Overall, this work provides a comprehensive map of variant effects for the PD domain of PAX6 and has provided evidence to enable clinical variant interpretation.

The key conclusions are convincing since variant scores were benchmarked against known pathogenic and benign variants, and variant scores agree with biochemical expectations. ClinVar classifies 46% of PAX6 missense variants as variants of unknown significance, which is an uninformative classification for patients and clinicians and informs the significance of this study. The work sufficiently meets these demands by providing context-based, functional evidence for PAX6 variants, advancing technically and clinically in the field. This work is significant because previously, no comprehensive functional variant evaluation had been done for PAX6. The audience that will be interested in this study are researchers and clinicians that are interested in diseases caused by PAX6 variants, and researchers interested in deep mutational scanning. I have no major concerns.

Minor Points:

- The abstract could mention the -Geneticin fitness assay
- Is this gene actionable? Would patients benefit from earlier diagnosis, and does it improve prognosis? If so, mentioning this could make the paper more significant.
- The GAL4A promoter was not mentioned in any of the text or figures, was the Geneticin driven by the GAL4 promoter when the AD was present?
- Out of interest, would you expect the assay to work had you included a larger bait sequence like an entire promoter that includes the bait sequence and not used a Y1H system?
- Why was CRY x3 eliminated as a DNA response element if your small-scale results in S1A suggest the assay will work with CRY?
- Highlight which amino acids are negatively charged on the map axis (triangles are present but this is not mentioned in text).
- Prolines also disrupt beta sheet structures, not just alpha helices, and it looks like this is true for both your maps, at least for beta sheet 2.
- More descriptive axis needed in figures: S2C needs 'fitness score', what does dataset1 and dataset2 mean in S2B (technical replicates?), S3 could benefit from n values above or below the violin plots, 2C and 5C need a y-axis (higher means higher conservation?).
- In-text mention of fitness score calculation outside of the methods was unclear: Was each time point compared to time point 0 to calculate the fitness score, and what was time point 0?
- Since the predictive ability of the map in Figure 5 is referenced in Figure 4, some sort of re-arranging could be done.
- The same plots for synonymous/nonsense and pathogenic/benign variant distributions shown in Figure 4A are needed for the - Geneticin assay.
- I would recommend if space is an issue that the explanations for the observations about the fitness assay were to be removed and only focus on the one theory that is most likely.
- This might be a file transfer issue, but the image files were pixelated, especially in 4c.
- In figure 5e, the correlation does not look linear. Do you have any thoughts as to why the intermediate-scoring variants (those in the bottom left of 5e, that are unable to upregulate geneticin, but have a growth phenotype closer to WT) correlate positively?

Reviewer #2:

The manuscript titled "Deep mutational scanning quantifies DNA binding and predicts clinical outcomes of PAX6 variants" details a novel application of the yeast one-hybrid (Y1H) assay to conduct deep mutational scanning (DMS) on the transcription factor (TF) protein PAX6. Y1H is commonly used to investigate putative TF binding sequences, but the authors instead fixed the

binding sequence and utilized Y1H to screen a mutational library of the TF PAX6. The authors created yeast cell lines with a PAX6 DNA response element leading to a geneticin resistance gene, and transformed the yeast with the mutational library. The yeasts were then treated with geneticin, and the surviving yeasts were sequenced using next-generation sequencing. The results from the DMS indicate that many of the amino acid residues are mutationally intolerant, especially those within alpha helices 3 and 6, which have been reported to directly bind to the DNA response element. Interestingly, the scanning results using the LE9 DNA response element are noticeably different in some key residues to that obtained from the BLX element, such as S54 and I71. The authors confirmed their findings with known protein structures, and determined that the residues closest to the DNA binding site tend to be more conserved and less mutationally tolerant.

The authors then utilized their findings to predict whether variants of PAX6 are benign or pathogenic. The predictions derived from the DMS dataset outperformed all existing computational methods based on variant effect predictors. Finally, the authors studied the interference that PAX6 expression generates against the natural proliferation of the host cells. The results suggest that PAX6 binds non-specifically to the yeast's genomic DNA and reduces growth. Thus the authors utilized yeast growth as a screening criterion and did another round of Y1H screening. The result matches with the expectation, in that the mutations that were detrimental to PAX6 DNA binding increases the growth rate of the host cells.

While the research and results presented in this manuscript are sound and convincing, we do believe that some major edits with regards to the structure and flow of the manuscript are required before the manuscript can be accepted.

Major concerns:

1. Main text. The introduction section ends with a general description of the experimental workflow, and the results section immediately jumps into the details of the results. It would be beneficial to add a paragraph before the first results section to introduce the experimental workflow in more detail.
2. Fig. 3c is not mentioned in the main text. Consider mentioning it at the start of the "Structural Sensitivity of PAX6 to Amino Acid Substitutions" section.
3. Fig. 4c left. The p-values are significant between some of the distributions, but the effect size is quite small overall. It would be beneficial to mention the effect size in the main text to provide a more complete picture of the finding.
4. Main text results section. The "PAX6 reduces yeast growth..." section should be introduced before the "DMS accurately classifies pathogenic variants..." section. Similarly, Fig. 5 should move to before Fig. 4. The current Fig. 5 provides context for the green bars in Fig. 4b, and it is confusing to the reader to see the current Fig. 4b first without knowing what the green bars correspond to, then read the current Fig. 5 to learn the context.
5. Fig. 5e. The correlation in the BLX assays has a fairly low R value, which indicates a weak negative correlation. This should be mentioned and possibly explained in the main text since the LE9 assays have a stronger correlation in comparison.
6. Fig. S1a. The highest concentration of geneticin is labeled as 8000 ug/ml. However, the concentration of geneticin used in S1b is only 800 ug/ml. Please double check the concentrations and make sure they are correct.
7. Fig. S1a. The CRYx3 DNA response element showed similar behaviors compared to LE9x3 and BLXx3. Why is it not studied? Please elaborate on the decision to drop CRYx3 in the supplementary material.
8. Fig. S1b. Arg92Glu still provided antibiotic resistance, despite it disrupting the DNA binding capacity of PAX6. Please elaborate on the inconsistency in the supplementary material.
9. Resolutions of main text figures are low. Panels such as Fig. 4b are almost unreadable. Please double check with the journal to make sure figures of sufficient resolutions are provided.

Minor concerns:

1. Fig. 2d. The two populations share very similar colors. Please consider using more distinct colors.
2. Fig. 3. It is somewhat confusing to refer to the center panels as "Fig. 3a central" and "Fig. 3b central". Labeling all panels with different letters may improve the clarity of the figure.

Reviewer #3:

In this manuscript, "Deep mutational scanning quantifies DNA binding and predicts clinical outcomes of PAX6 variants", the authors present a new clinically-relevant dataset describing the impact of thousands of amino acid variants in the transcription factor PAX6. Overall, this is a very nice paper that presents an approach that could be employed across diverse transcription factors, but it needs a more thorough discussion of the importance of the promiscuous DNA binding result.

Major Comments

1 I was unable to find the actual data from the study. Figures 2 and 5 include sequence-function maps, but the data plotted are not included. The authors should submit the counts and scores for each variant to MaveDB in order to share the data with the community as a condition of publication and include the relevant accession numbers in the manuscript. The authors should reach out to the MaveDB maintainers if they need assistance converting the Enrich2 files to a suitable format. The authors should also upload their raw sequencing data to GEO and include the barcode-variant map as an additional file so that other researchers can reanalyze the data if desired. The lack of any data availability statement or section of the manuscript strikes me as a glaring oversight and I look forward to it being corrected.

2 The "inverse" result described when growing the cells in the absence of Geneticin is remarkable and deserves more discussion. While I'm not going to ask the authors to perform additional experiments, there are quite a few possible hypotheses here that could be explored, and I think that the way this result is somewhat buried in the text is disingenuous. This seems to suggest that most of the effect of PAX6 variants is that they interfere with any DNA binding rather than being PAX6-specific. Does that seem to be the case? Are there variants that allow growth both with and without Geneticin, suggesting that they are highly specific for the LE9 or BLX element? Does this suggest that PAX6 binding is (in general) not very specific or that low-efficiency off-target binding is sufficient to kill yeast? In an earlier section of the manuscript, the authors describe the differences between LE9 and BLX datasets and call out specific variants that seem to be sequence specific. I'd like to see these sections brought closer together or at least discussed in a more holistic way.

3 I'd like to thank the authors for including so much detail in the experimental methods. It was very easy to follow.

Minor Comments

1 In the introduction paragraph 4 sentence 1, the authors should consider also citing the highly relevant and more recent review by Tabet et al. (PMID 36055970).

2 In the first paragraph of the results, the authors mention the 14 candidate DNA response elements that were rejected. These appear to be included in Table S1 but there's no callout to the table here.

3 The Table S1 legend refers to row numbers but there are no numbers on the table itself. Adding row numbers would make this much easier to navigate.

4 Control variants (variants with known effect) that were used to calibrate the assay should be listed clearly in a supplemental table.

5 The human pathogenic variants mentioned in the "Human variants and phenotypes" section should be listed, along with their source, group, and classification, in a supplemental table.

6 The barcode-based analysis described in the first paragraph of the section "PAX6 variants cause sequence-specific perturbations in DNA-binding" is interesting. It would be nice to see the distribution of barcode scores as a supplemental figure, as well as the result of multiple replicates of this splitting analysis.

7 In the last paragraph of the section "PAX6 variants cause sequence-specific perturbations in DNA-binding", the authors mention specific variants that seem to have opposing effects on fitness. Do these variants contact the DNA directly? It would be useful to provide some additional context when these are discussed.

8 At the end of the first paragraph of the section "Structural Sensitivity of PAX6 to Amino Acid Substitutions", the authors mention residues having "the lowest scores". Is this the lowest average score? Median score?

9 The capitalization of the " Structural Sensitivity of PAX6 to Amino Acid Substitutions" section title is inconsistent with the others.

10 In the first sentence of the section "DMS accurately classifies pathogenic variants and gives insight into disease phenotypes", the authors should also cite Fayer et al. (PMID 34793697).

11 The second sentence of the section "DMS accurately classifies pathogenic variants and gives insight into disease phenotypes" might be better phrased as "... PAX6 fitness scores could help explain variant pathogenicity" as it's sometimes frowned upon to refer to human variants as "mutations".

12 The authors should provide a supplemental table with the gnomAD variants mentioned in the first paragraph of the section "DMS accurately classifies pathogenic variants and gives insight into disease phenotypes".

13 In the section "DMS accurately classifies pathogenic variants and gives insight into disease phenotypes", the authors list two p-values in the fourth paragraph but don't state what test was used.

14 In the second paragraph of the section "DMS accurately classifies pathogenic variants and gives insight into disease phenotypes", the authors should list and cite all the predictors used in the study.

15 The authors should provide the evidence assigned to each variant in the study. This could be included as part of the upload to MaveDB as requested above, or provided as an additional supplemental table.

16 In the section "PAX6 reduces yeast growth through promiscuous binding to the genome" the authors should consider mentioning and citing the related paper on fitness without antibiotic selection by Mehlhoff et al. (PMID 32385156).

17 In the discussion, the authors mention the PAX6(5a) alternate transcript. Are there sufficient clinical variants in PAX6(5a) that the authors can comment on the utility of this PAX6 dataset to interpret those variants?

18 The authors should mention new methods such as PrimateAI and AlphaMissense in the discussion where the other predictors are being discussed, but I don't think it's required that they compare their dataset to these methods as they are quite new and still being critically evaluated by the field. I'm fine with this being described as "future work".

19 In the methods, the authors should include an appropriate citation for the False Discovery Rate calculation in the second paragraph in the section "Barcode sequencing and fitness score calculations".

20 In the methods in the first sentence of the section "Structural analysis of fitness scores", where did the annotation ranges come from? Was this from UniProt, the authors own knowledge, or other?

21 The last sentence of the section "Structural analysis of fitness scores" in the methods is missing a citation for PyMol.

22 The first sentence of the section "Human variants and phenotypes" in the methods is missing citations for HGUeye, ClinVar, and HGMD.

23 Again in the section "Human variants and phenotypes", the authors should include OMIM or MONDO terms for the conditions mentioned.

24 In the methods section titled "VEP benchmarking", the methods listed in the third and fourth sentences are not accompanied by citations.

25 Again in the "VEP benchmarking" section, a method called "Eigen" is mentioned but It's not included in the previous lists or cited.

Data accession links

Raw and processed sequencing data on GEO:

<https://www.ncbi.nlm.nih.gov/geo/query/acc.cgi?acc=GSE253580>

GEO reviewer token:

klafkimojtonpiv

MAVE-DB accession:

<https://www.mavedb.org/#/experiments/urn:mavedb:00000665-a>

Reviewer #1:

McDonnell et. al present an evaluation of missense variant effects of the PD domain of the PAX6 transcription factor in three unique contexts, including the evaluation of PAX6 binding to two different DNA elements (LE9 and BLX) via yeast-1-hybrid assays, and a fitness assay. First, the barcoded variant library of the PD domain was generated in-house using PCR-based saturation mutagenesis. Next, deep mutational scanning was used to evaluate variants in high-throughput, selectable yeast systems, either by functional variants driving antibiotic resistance, or via fitness. Barcode sequencing and analysis of the allele frequencies over time points generated the sequence-function maps. The results agreed with biochemical expectations, considering that the nonsense and synonymous variants were well-separated in the assays and that deleterious variant scores were found in DNA-proximal positions. Known benign and pathogenic variants of PAX6 performed as expected in the assays. The functional scores of some variant positions differed between the two Y1H assays, which was likely due to differences in DNA-binding specificity. All assays performed better or on par with computational predictors of variant effects, with the best performer being the BLX fitness assay. The sequence-function map was able to provide support for over 2000 missense variants to more clinically-useful classifications such as likely benign or likely pathogenic, according to ACMG guidelines. Overall, this work provides a comprehensive map of variant effects for the PD domain of PAX6 and has provided evidence to enable clinical variant interpretation.

The key conclusions are convincing since variant scores were benchmarked against known pathogenic and benign variants, and variant scores agree with biochemical expectations. ClinVar classifies 46% of PAX6 missense variants as variants of unknown significance, which is an uninformative classification for patients and clinicians and informs the significance of this study. The work sufficiently meets these demands by providing context-based, functional evidence for PAX6 variants, advancing technically and clinically in the field. This work is significant because previously, no comprehensive functional variant evaluation had been done for PAX6. The audience that will be interested in this study are researchers and clinicians that are interested in diseases caused by PAX6 variants, and researchers interested in deep mutational scanning. I have no major concerns.

Minor Points:

- The abstract could mention the -Geneticin fitness assay

Done.

- Is this gene actionable? Would patients benefit from earlier diagnosis, and does it improve prognosis? If so, mentioning this could make the paper more significant.

Now mentioned in the introduction: “Knowledge of the impact of PAX6 missense variants on binding to DNA is key for genetic diagnosis of eye development disorders and can facilitate patient care^{28,29}.”

- The GAL4A promoter was not mentioned in any of the text or figures, was the Geneticin driven by the GAL4 promoter when the AD was present?

There is no GAL4 promoter in this study – the only GAL4-derived element we use is the GAL4 activation domain (GAL4AD), which is translationally fused to PAX6 in our constructs. The activation domain contains a nine amino acid Transactivation Domain (9aaTAD) that interacts with components of the transcription machinery. The expression of GeneticinR gene is driven by binding of PAX6-GAL4AD fusion to a PAX6-binding site. We clarified this in Figure 1.

- Out of interest, would you expect the assay to work had you included a larger bait sequence like an entire promoter that includes the bait sequence and not used a Y1H system?

The bait sequences we tested ranged from 12 nt (BLX x1) to 250 nt (tfap2a) in length. Those that worked best had intermediate lengths (36 nt, BLX x3 and 141 nt, LE9 x3), but this is based on a small number of baits tested. Given the mechanism of GAL4 transactivation, we expect that it should be possible to use an entire promoter rather than repeats of binding sites in this assay, though we have not tested this.

- Why was CRY x3 eliminated as a DNA response element if your small-scale results in S1A suggest the assay will work with CRY?

Although initial spot assays suggested that the CRY x3 strain might be suitable, follow-up experiments in liquid media showed significant growth of this strain in high concentrations of geneticin (800 ug/uL) in the absence of PAX6. This suggested that the effects of PAX6 mutations may be difficult to interpret and we decided not to pursue this strain further. We included this information in the results section.

- Highlight which amino acids are negatively charged on the map axis (triangles are present but this is not mentioned in text).

Done in the figure legend.

- Prolines also disrupt beta sheet structures, not just alpha helices, and it looks like this is true for both your maps, at least for beta sheet 2.

Now corrected in the results section.

- More descriptive axis needed in figures: S2C needs 'fitness score', what does dataset1 and dataset2 mean in S2B (technical replicates?), S3 could benefit from n values above or below the violin plots, 2C and 5C need a y-axis (higher means higher conservation?).

Done.

- In-text mention of fitness score calculation outside of the methods was unclear: Was each time point compared to time point 0 to calculate the fitness score, and what was time point 0?

The first four time points of each experiment were used (T=0, 12, 24 and 36 h). T=0 represents the time just before antibiotic was added to the media and the first sample was collected. We clarified this in the methods section.

- Since the predictive ability of the map in Figure 5 is referenced in Figure 4, some sort of re-arranging could be done.

Done. Figures 4 and 5 have been swapped.

- The same plots for synonymous/nonsense and pathogenic/benign variant distributions shown in Figure 4A are needed for the -Geneticin assay.

Done.

- I would recommend if space is an issue that the explanations for the observations about the fitness assay were to be removed and only focus on the one theory that is most likely.

We kept these explanation as they might appear as plausible alternatives to some readers.

- This might be a file transfer issue, but the image files were pixelated, especially in 4c.

Corrected.

- In figure 5e, the correlation does not look linear. Do you have any thoughts as to why the intermediate-scoring variants (those in the bottom left of 5e, that are unable to upregulate geneticin, but have a growth phenotype closer to WT) correlate positively?

We added some thoughts in the discussion about the possible origin of the nonlinear negative correlation. We speculate that the nonlinearity results from an interplay between the growth promoting effects of PAX6 through expression of the antibiotic resistance gene, and growth reducing effects caused by promiscuous binding to yeast genomic DNA. The relative strength of the two effects may differ between variants, for example some variants may lose binding to the bait sequence, resulting in low growth with antibiotic, but retain promiscuous binding to the yeast genome, resulting in low growth without antibiotic. Further studies are needed to deconvolve these effects, perhaps by exposing yeast to a range of antibiotic concentrations which could result in the pattern shown below:

Reviewer #2:

The manuscript titled "Deep mutational scanning quantifies DNA binding and predicts clinical outcomes of PAX6 variants" details a novel application of the yeast one-hybrid (Y1H) assay to conduct deep mutational scanning (DMS) on the transcription factor (TF) protein PAX6. Y1H is commonly used to investigate putative TF binding sequences, but the authors instead fixed the binding sequence and utilized Y1H to screen a mutational library of the TF PAX6. The authors created yeast cell lines with a PAX6 DNA response element leading to a geneticin resistance gene, and transformed the yeast with the mutational library. The yeasts were then treated with geneticin, and the surviving yeasts were sequenced using next-generation sequencing.

The results from the DMS indicate that many of the amino acid residues are mutationally intolerant, especially those within alpha helices 3 and 6, which have been reported to directly bind to the DNA response element. Interestingly, the scanning results using the LE9 DNA response element are noticeably different in some key residues to that obtained from the BLX element, such as S54 and I71. The authors confirmed their findings with known protein structures, and determined that the residues closest to the DNA binding site tend to be more conserved and less mutationally tolerant.

The authors then utilized their findings to predict whether variants of PAX6 are benign or pathogenic. The predictions derived from the DMS dataset outperformed all existing computational methods based on variant effect predictors. Finally, the authors studied the interference that PAX6 expression generates against the natural proliferation of the host cells. The results suggest that PAX6 binds non-specifically to the yeast's genomic DNA and reduces growth. Thus the authors utilized yeast growth as a screening criterion and did another round of Y1H screening. The result matches with the expectation, in that the mutations that were detrimental to PAX6 DNA binding increases the growth rate of the host cells.

While the research and results presented in this manuscript are sound and convincing, we do believe that some major edits with regards to the structure and flow of the manuscript are required before the manuscript can be accepted.

Major concerns:

1. Main text. The introduction section ends with a general description of the experimental workflow, and the results section immediately jumps into the details of the results. It would be beneficial to add a paragraph before the first results section to introduce the experimental workflow in more detail.

Done. We added the following text at the beginning of the results section:

“We designed an experimental strategy to systematically measure the effects of single amino acid mutations in the human *PAX6* paired domain on binding to DNA. The key steps of our approach are: (1) the construction of a saturation mutagenesis library of *PAX6*; (2) expression of *PAX6* variants in a yeast strain containing a *PAX6* binding site upstream of an antibiotic resistance gene; (3) pooled competitive growth assays in the presence and absence of antibiotic; (4) high-throughput sequencing of samples isolated at multiple time points during competitive growth to associate functional scores with *PAX6* variants.”

2. Fig. 3c is not mentioned in the main text. Consider mentioning it at the start of the "Structural Sensitivity of *PAX6* to Amino Acid Substitutions" section.

Done.

3. Fig. 4c left. The p-values are significant between some of the distributions, but the effect size is quite small overall. It would be beneficial to mention the effect size in the main text to provide a more complete picture of the finding.

We now conclude this paragraph by saying that the effect sizes are small but statistically significant.

4. Main text results section. The "PAX6 reduces yeast growth..." section should be introduced before the "DMS accurately classifies pathogenic variants..." section. Similarly, Fig. 5 should move to before Fig. 4. The current Fig. 5 provides context for the green bars in Fig. 4b, and it is confusing to the reader to see the current Fig. 4b first without knowing what the green bars correspond to, then read the current Fig. 5 to learn the context.

Done.

5. Fig. 5e. The correlation in the BLX assays has a fairly low R value, which indicates a weak negative correlation. This should be mentioned and possibly explained in the main text since the LE9 assays have a stronger correlation in comparison.

We added some thoughts in the discussion about the interpretation of these correlations. We speculate that the nonlinearity results from an interplay between the growth promoting effects of *PAX6* through expression of the antibiotic resistance gene, and growth reducing effects caused by promiscuous binding to yeast genomic DNA. The relative strength of the two effects may differ between variants and between strains; for example, some variants may lose binding to the bait sequence, resulting in low growth with antibiotic, but retain promiscuous binding to the yeast genome, resulting in low growth without antibiotic. We refrained from labelling correlations as either weak or strong and left this judgement to the reader.

6. Fig. S1a. The highest concentration of geneticin is labeled as 8000 ug/ml. However, the concentration of geneticin used in S1b is only 800 ug/ml. Please double check the concentrations and make sure they are correct.

Corrected.

7. Fig. S1a. The CRYx3 DNA response element showed similar behaviors compared to LE9x3 and BLXx3. Why is it not studied? Please elaborate on the decision to drop CRYx3 in the supplementary material.

Although initial spot assays suggested that the CRY x3 strain might be suitable, follow-up experiments in liquid media showed significant growth of this strain in high concentrations of geneticin (800 ug/uL) in the absence of PAX6. This suggested that the effects of PAX6 mutations may be difficult to interpret and we decided not to pursue this strain further. We included this information in the results section.

8. Fig. S1b. Arg92Glu still provided antibiotic resistance, despite it disrupting the DNA binding capacity of PAX6. Please elaborate on the inconsistency in the supplementary material.

Variants Cys52Arg (aniridia) and Ser54Arg (non-aniridia) previously showed substantially perturbed binding to LE9 (<https://www.ncbi.nlm.nih.gov/pmc/articles/PMC7056646/>), while Arg92Gln (benign) showed milder disruption to LE9 binding. These results are recapitulated in our Fig S1b. We added this information to the figure legend and supplementary dataset.

9. Resolutions of main text figures are low. Panels such as Fig. 4b are almost unreadable. Please double check with the journal to make sure figures of sufficient resolutions are provided.

Corrected.

Minor concerns:

1. Fig. 2d. The two populations share very similar colors. Please consider using more distinct colors.

Done.

2. Fig. 3. It is somewhat confusing to refer to the center panels as "Fig. 3a central" and "Fig. 3b central". Labeling all panels with different letters may improve the clarity of the figure.

Done.

Reviewer #3:

In this manuscript, "Deep mutational scanning quantifies DNA binding and predicts clinical outcomes of PAX6 variants", the authors present a new clinically-relevant dataset describing the impact of thousands of amino acid variants in the transcription factor PAX6. Overall, this is a very nice paper that presents an approach that could be employed across diverse transcription factors, but it needs a more thorough discussion of the importance of the promiscuous DNA binding result.

Major Comments

1 I was unable to find the actual data from the study. Figures 2 and 5 include sequence-function maps, but the data plotted are not included. The authors should submit the counts and scores for each variant to MaveDB in order to share the data with the community as a condition of publication and include the relevant accession numbers in the manuscript. The authors should reach out to the MaveDB maintainers if they need assistance converting the Enrich2 files to a suitable format. The authors should also upload their raw sequencing data to GEO and include the barcode-variant map as an additional file so that other researchers can reanalyze the data if desired. The lack of any data availability statement or section of the manuscript strikes me as a glaring oversight and I look forward to it being corrected.

We uploaded to GEO: the raw barcode sequencing files, barcode counts produced by Enrich2, barcode-variant map raw data from PacBio and processed barcode maps. The GEO accession number is GSE253580. We also uploaded variant scores and annotations to MaveDB (four data series, accession urn:mavedb:00000665-a). We added these accession numbers to the manuscript.

2 The "inverse" result described when growing the cells in the absence of Geneticin is remarkable and deserves more discussion. While I'm not going to ask the authors to perform additional experiments, there are quite a few possible hypotheses here that could be explored, and I think that the way this result is somewhat buried in the text is disingenuous.

Following this comment and those of the other reviewers, we have now included these results in the abstract and we reordered figures and restructured the results section to give more prominence to this finding.

This seems to suggest that most of the effect of PAX6 variants is that they interfere with any DNA binding rather than being PAX6-specific. Does that seem to be the case?

Indeed, most variants that perturb DNA-binding do so in all four conditions: LE9 and BLX, with and without geneticin. This seems to suggest that single mutations are more likely to influence the strength of DNA-binding in general, rather than influence sequence specificity. However, we also found multiple examples of individual mutations that changed sequence-specific binding. We now say this explicitly in the discussion.

Are there variants that allow growth both with and without Geneticin, suggesting that they are highly specific for the LE9 or BLX element? Does this suggest that PAX6 binding is (in general) not very specific or that low-efficiency off-target binding is sufficient to kill yeast? In an earlier section of the manuscript, the authors describe the differences between LE9 and BLX datasets and call out specific variants that seem to be sequence specific. I'd like to see these sections brought closer together or at least discussed in a more holistic way.

Thanks for these suggestions. We brought the two results sections next to each other (as also suggested by the other reviewers) and we expanded the discussion to address these points. With regard to binding specificity towards LE9 vs BLX elements, we believe that a direct comparison of experiments in presence of geneticin is most interpretable, and so we

base our discussion of specificity upon experiments with geneticin. Also, we do not mean to say that low efficiency off-target binding will kill yeast: yeast cells expressing wild-type PAX6 are alive, they just grow slower than cells with inactive PAX6 variants.

3 I'd like to thank the authors for including so much detail in the experimental methods. It was very easy to follow.

Thanks.

Minor Comments

1 In the introduction paragraph 4 sentence 1, the authors should consider also citing the highly relevant and more recent review by Tabet et al. (PMID 36055970).

Done.

2 In the first paragraph of the results, the authors mention the 14 candidate DNA response elements that were rejected. These appear to be included in Table S1 but there's no callout to the table here.

Corrected.

3 The Table S1 legend refers to row numbers but there are no numbers on the table itself. Adding row numbers would make this much easier to navigate.

Corrected.

4 Control variants (variants with known effect) that were used to calibrate the assay should be listed clearly in a supplemental table.

Done in the new Dataset S1.

5 The human pathogenic variants mentioned in the "Human variants and phenotypes" section should be listed, along with their source, group, and classification, in a supplemental table.

Done in the new Dataset S1.

6 The barcode-based analysis described in the first paragraph of the section "PAX6 variants cause sequence-specific perturbations in DNA-binding" is interesting. It would be nice to see the distribution of barcode scores as a supplemental figure, as well as the result of multiple replicates of this splitting analysis.

The distribution of barcode scores is nearly indistinguishable from the distribution of variant scores, so showing this does not add much information. We show replicates of the splitting analysis in the new Fig S3c-d.

7 In the last paragraph of the section "PAX6 variants cause sequence-specific perturbations in DNA-binding", the authors mention specific variants that seem to have opposing effects on fitness. Do these variants contact the DNA directly? It would be useful to provide some additional context when these are discussed.

All these variants are within 5 Å from the nearest non-hydrogen atom in DNA, which we now say in the text.

8 At the end of the first paragraph of the section "Structural Sensitivity of PAX6 to Amino Acid Substitutions", the authors mention residues having "the lowest scores". Is this the lowest average score? Median score?

We meant the lowest median scores; we've now added this information in the text.

9 The capitalization of the " Structural Sensitivity of PAX6 to Amino Acid Substitutions" section title is inconsistent with the others.

Corrected.

10 In the first sentence of the section "DMS accurately classifies pathogenic variants and gives insight into disease phenotypes", the authors should also cite Fayer et al. (PMID 34793697).

Done.

11 The second sentence of the section "DMS accurately classifies pathogenic variants and gives insight into disease phenotypes" might be better phrased as "... PAX6 fitness scores could help explain variant pathogenicity" as it's sometimes frowned upon to refer to human variants as "mutations".

Corrected.

12 The authors should provide a supplemental table with the gnomAD variants mentioned in the first paragraph of the section "DMS accurately classifies pathogenic variants and gives insight into disease phenotypes".

Done in the new Dataset S1.

13 In the section "DMS accurately classifies pathogenic variants and gives insight into disease phenotypes", the authors list two p-values in the fourth paragraph but don't state what test was used.

We've used the Wilcoxon test with Benjamini-Hochberg (FDR) correction for multiple testing, we now say so in the text.

14 In the second paragraph of the section "DMS accurately classifies pathogenic variants

and gives insight into disease phenotypes", the authors should list and cite all the predictors used in the study.

We have included a supplementary table with a list of all predictors including citations.

15 The authors should provide the evidence assigned to each variant in the study. This could be included as part of the upload to MaveDB as requested above, or provided as an additional supplemental table.

Done.

16 In the section "PAX6 reduces yeast growth through promiscuous binding to the genome" the authors should consider mentioning and citing the related paper on fitness without antibiotic selection by Mehlhoff et al. (PMID 32385156).

Done in the discussion section "Effect of PAX6 in the absence of antibiotic". Thanks for bringing our attention to this very relevant paper!

17 In the discussion, the authors mention the PAX6(5a) alternate transcript. Are there sufficient clinical variants in PAX6(5a) that the authors can comment on the utility of this PAX6 dataset to interpret those variants?

Literature suggests that the 14aa insertion in the PAX6(5a) splice variant causes a shift in binding contributions made by the NTS/ CTS when interacting with DNA. In canonical PAX6, the NTS is primarily used, while in PAX6(5a) CTS seems to be more important. We don't know how well our data would predict the effects of mutations on PAX6(5a):DNA binding. However, the fact that our data correlates well with clinical outcomes suggests that either canonical PAX6 is more important clinically, or that the effects of mutations in both variants are similar.

18 The authors should mention new methods such as PrimateAI and AlphaMissense in the discussion where the other predictors are being discussed, but I don't think it's required that they compare their dataset to these methods as they are quite new and still being critically evaluated by the field. I'm fine with this being described as "future work".

We had already included PrimateAI in the analysis (Table S2), it performed surprisingly poorly. AlphaMissense performed better but was not the top predictor for PAX6; we refrained from adding it to the paper or singling it out in the discussion.

19 In the methods, the authors should include an appropriate citation for the Falst Discovery Rate calculation in the second paragraph in the section "Barcode sequencing and fitness score calculations".

Done.

20 In the methods in the first sentence of the section "Structural analysis of fitness scores",

where did the annotation ranges come from? Was this from UniProt, the authors own knowledge, or other?

This was done according to the crystal structure, now cited.

21 The last sentence of the section "Structural analysis of fitness scores" in the methods is missing a citation for PyMol.

Done.

22 The first sentence of the section "Human variants and phenotypes" in the methods is missing citations for HGUeye, ClinVar, and HGMD.

We provided a reference to a supplementary table instead, as we noticed that the HGUeye database is not currently available online.

23 Again in the section "Human variants and phenotypes", the authors should include OMIM or MONDO terms for the conditions mentioned.

We have not used OMIM or MONDO terms, instead the phenotypes were defined based on Williamson et al. (Genet Med 2020).

24 In the methods section titled "VEP benchmarking", the methods listed in the third and fourth sentences are not accompanied by citations.

All VEPs are now listed and cited in Dataset S4.

25 Again in the "VEP benchmarking" section, a method called "Eigen" is mentioned but it's not included in the previous lists or cited.

All VEPs are now listed and cited in Dataset S4.

1st Mar 2024

Manuscript Number: MSB-2023-11985R

Title: Deep mutational scanning quantifies DNA binding and predicts clinical outcomes of PAX6 variants

Dear Dr. Kudla,

Thank you for the submission of your revised manuscript to Molecular Systems Biology. We have now received the enclosed reports from the referees that were asked to re-assess it. As you will see the reviewers are now globally supportive and I am pleased to inform you that we will be able to accept your manuscript pending the following final amendments and appropriate response to reviewers:

- 1) As part of Reviewer 3's remaining concerns, they have requested that the citations for GEO (PMID 11752295) and MaveDB (PMID 31679514) be added to the Data Availability section. At MSB we restrict the datasets listed in the Data Availability statement to those newly generated in the current study. However, previously published datasets that have been analyzed in the study should be indicated with accession numbers and URLs in either a separate section of the Materials and Methods called 'Reanalyzed datasets' or within other Materials and Methods sections that explain the analysis of the previous datasets.
- 2) Please note that we require an institutional email address for the corresponding author in the manuscript file and in our electronic submission system.
- 3) The corresponding author should be indicated in the main manuscript file along with an institutional email address as a contact.
- 4) In the main manuscript file, please do the following:
 - Add up to 5 keywords.
 - Please format the Data availability section according to the example below (please be sure to include the relevant URLs):

The datasets and computer code produced in this study are available in the following databases:

- Chip-Seq data: Gene Expression Omnibus GSE46748 (<https://www.ncbi.nlm.nih.gov/geo/query/acc.cgi?acc=GSE46748>)
 - Modeling computer scripts: GitHub (<https://github.com/SysBioChalmers/GECKO/releases/tag/v1.0>)
 - [data type]: [full name of the resource] [accession number/identifier] ([doi or URL or identifiers.org/DATABASE:ACCESSION])
 - Data availability: The GSE253580 dataset is not publicly available. Please be aware that all deposited datasets must be freely accessible for acceptance.
 - Please add a "Disclosure and competing interests statement". We updated our journal's competing interests policy in January 2022 and request authors to consider both actual and perceived competing interests. Please review the policy <https://www.embopress.org/competing-interests> and update your competing interests if necessary.
 - Please correct the reference citation in the reference list. The references should be alphabetical, not numerical. Where there are more than 10 authors on a paper, note that only 10 should be listed, followed by "et al.". Please check "Author Guidelines" for more information.
<https://www.embopress.org/page/journal/17574684/authorguide#referencesformat>
- 5) Please place individual sections of the manuscript in the following order: Title page - Abstract & Keywords - Introduction - Results - Discussion - Materials and Methods - Data Availability - Acknowledgements - Disclosure and Competing Interests Statement - References - Figure Legends - Expanded View Figure Legends - Tables.
 - 6) Please rename "Methods" to "Materials and Methods"
 - 7) For the figures and figure legends, please take care of the following:
 - Please make sure to update the callouts of all figures in the main manuscript text (currently figure callouts are missing for Figure 2d, 3adg, 4ac)
 - Please note that the box plots need to be defined in terms of minima, maxima, centre, bounds of box and whiskers, and percentile in the legends of figures 3a-f; 5f-g.
 - Please note that information related to n is missing in the legends of Figures 3a-f; 5f-g.
 - Please note that in figures 3a, c-d, f there is a mismatch between the annotated p values in the figure legend and the annotated p values in the figure file that should be corrected. Please note that we require exact p-values to be reported - currently these are missing.
 - For the Datasets EV2 and 3, as they are txt files, each of those will need its own separate README file and will need to be uploaded in a zipped folder - e.g. a Dataset EV2 zipped folder containing Dataset EV2 and its README file.
 - For the Dataset EV4, although the table is larger than 1 page, please include this as as Appendix Table S3 in the Appendix and include the references in the Appendix reference list. Please ensure there is a legend included and that the callout is updated in the main manuscript text (suggested in the Materials and Methods).
 - We would suggest that each EV figure be limited to a single page for ease of viewing for readers.
 - 8) Appendix file:
 - Please ensure that the updated file with the corrections to the Appendix Figure S1 is uploaded in the system.
 - In the Appendix file, please ensure the word "Appendix" is included in all labels for Appendix Figures and Appendix Tables (e.g.

Appendix Figure S1).

- Please add a Table of Contents to the Appendix with page numbers and ensure that the word "Appendix" is included for each table and figure.

- Please rename the Supplementary References to Appendix References. As in the main manuscript file, the Appendix References should be alphabetical, not numerical. Where there are more than 10 authors on a paper, note that only 10 should be listed, followed by "et al."

9) Please ensure that all funding sources are entered into the manuscript submission system (i.e. currently Precision Medicine PhD programme at the University of Edinburgh is missing)

10) Synopsis:

- Synopsis image: Please include a synopsis image that summarises the main findings of the manuscript on a glance as a high-resolution jpeg file 550 pixels wide x (250-400) pixels high.

- Synopsis text: Please provide a short standfirst (maximum of 300 characters, including space), limit the bullet points to max. 5 and upload it as a separate .doc file. Please write the bullet points to summarise the key NEW findings. They should be designed to be complementary to the abstract - i.e. not repeat the same text. We encourage inclusion of key acronyms and quantitative information (maximum of 30 words / bullet point). Please use the passive voice.

11) Source Data: Please ensure that a completed Source Data checklist is uploaded (as sent to you previously by Hannah Sonntag), along with a single source data file (zipped) per figure, with the panels clearly visible in the folder structure. Currently we are not able to open the .tsv file that has been uploaded.

12) As part of the EMBO Publications transparent editorial process initiative (see our policy here: https://www.embopress.org/transparent-process#Review_Process), Molecular Systems Biology will publish online a Peer Review File (PRF) to accompany accepted manuscripts. This file will be published in conjunction with your paper and will include the anonymous referee reports, your point-by-point response and all pertinent correspondence relating to the manuscript. Let us know whether you agree with the publication of the PRF and as here, if you want to remove or not any figures from it prior to publication. Please note that the Authors checklist will be published at the end of the PRF.

13) Please provide a point-by-point letter INCLUDING my comments as well as the reviewer's reports and your detailed responses (as Word file).

I look forward to reading a new revised version of your manuscript as soon as possible.

Yours sincerely,

Poonam Bheda, PhD
Scientific Editor
Molecular Systems Biology

Please click on the link below to submit your revised paper.

Reviewer #1:

Previous comments have been sufficiently addressed.

Reviewer #2:

Authors have successfully addressed all my comments.

Reviewer #3:

Thank you to the authors for their thorough and thoughtful response to the reviews. All of my previous comments and concerns have been fully addressed.

My one remaining request is that the authors add citations to the main text for all the predictors shown in Fig 5e, as well as

citations for GEO (PMID 11752295) and MaveDB (PMID 31679514) in the Data Availability section. Citation counts are critically important for demonstrating the utility of computational tools and resources, which is needed to help sustain these projects.

- Please format the Data availability section according to the example below (please be sure to include the relevant URLs):

"The datasets and computer code produced in this study are available in the following databases:

- Chip-Seq data: Gene Expression Omnibus GSE46748

(<https://www.ncbi.nlm.nih.gov/geo/query/acc.cgi?acc=GSE46748>)"

Done.

- Please double check the exact p-values listed in Figure 3a, c-d, f, as all 4 of the significant values are shown as $<2.2e-16$.

These are the only p-values we have; my understanding is that this is caused by the representation of floating-point numbers in the program that calculates p-values. Whenever the value is smaller than $2.2e-16$, the program does not calculate the exact value, it just reports it as smaller than $2.2e-16$.

- Please remove the (a) from Figure EV1, the (b) from Figure EV2, etc

Done.

- Please upload a jpeg (not PDF) of the synopsis image and ensure that the figure is high-resolution with a size of 550 pixels wide x (250-400) pixels high. We would also suggest simplifying the figure, as with the small size of 550 pixels wide the text will be difficult to read. For this we would suggest removing the bullet points, as this synopsis figure will be placed next to the synopsis text that already includes bullet points.

Done.

- For the synopsis text, please ensure a short standfirst (i.e. single general sentence of maximum of 300 characters, including spaces) is included before the bullet points. Please use the passive voice for this sentence.

Done.

- Please organise the Source Data so that there is a single zipped folder per figure and within this folder there should be a single file for each panel. All files and folders should be clearly labeled with the correct figure and figure panel.

Done.

- Please note that our procedure dictates that the authors need to have the institutional email address in the system while we can insert both institutional and personal in the manuscript file, so please add your current institutional email address to the manuscript file, and we will be happy to update your account in our online system for you.

Done. I've added the institutional address to the manuscript text, however I would like to keep my personal gmail address in the published version.

14th May 2024

Manuscript number: MSB-2023-11985RR

Title: Deep mutational scanning quantifies DNA binding and predicts clinical outcomes of PAX6 variants

Dear Dr. Kudla,

Congratulations on an excellent manuscript, I am pleased to inform you that your manuscript has been accepted for publication in Molecular Systems Biology. It has been a pleasure to work with you to get this to the acceptance stage.

Yours sincerely,

Poonam Bheda, PhD
Scientific Editor
Molecular Systems Biology
